# The vacuolar-ATPase complex and assembly factors, TMEM199 and CCDC115, control HIF1α prolyl hydroxylation by regulating cellular iron levels

Anna L Miles[†], Stephen P Burr[†], Guinevere L Grice, James A Nathan*

Department of Medicine, Cambridge Institute for Medical Research, University of Cambridge, Cambridge, United Kingdom

**Abstract** Hypoxia Inducible transcription Factors (HIFs) are principally regulated by the 2-oxoglutarate and Iron(II) prolyl hydroxylase (PHD) enzymes, which hydroxylate the HIFα subunit, facilitating its proteasome-mediated degradation. Observations that HIFα hydroxylation can be impaired even when oxygen is sufficient emphasise the importance of understanding the complex nature of PHD regulation. Here, we use an unbiased genome-wide genetic screen in near-haploid human cells to uncover cellular processes that regulate HIF1α. We identify that genetic disruption of the Vacuolar H+ ATPase (V-ATPase), the key proton pump for endo-lysosomal acidification, and two previously uncharacterised V-ATPase assembly factors, TMEM199 and CCDC115, stabilise HIF1α in aerobic conditions. Rather than preventing the lysosomal degradation of HIF1α, disrupting the V-ATPase results in intracellular iron depletion, thereby impairing PHD activity and leading to HIF activation. Iron supplementation directly restores PHD catalytic activity following V-ATPase inhibition, revealing important links between the V-ATPase, iron metabolism and HIFs.

*For correspondence: jan33@cam.ac.uk

[†]These authors contributed equally to this work

**Competing interests:** The authors declare that no competing interests exist.

## Introduction

HIFs are major transcriptional regulators of cellular responses to oxygen availability, promoting several metabolic adaptations to ensure cell survival. In aerobic conditions, the HIFα subunit is constitutively expressed but rapidly degraded by the proteasome, in a process requiring two post-translational modifications: (i) prolyl hydroxylation of the HIFα oxygen dependent degradation (ODD) domain by prolyl hydroxylases (PHDs) (*Bruick and McKnight, 2001*; *Epstein et al., 2001*), and (ii) subsequent ubiquitination by the von-hippel lindau (VHL) E3 ligase (*Maxwell et al., 1999*). Prolyl hydroxylation of HIFα acts as the recruitment signal for VHL, which rapidly ubiquitinates the ODD domain facilitating proteasomal degradation. Indeed, HIF1α (the ubiquitously expressed HIFα isoform) is a very short-lived protein (*Berra et al., 2001*), and the efficiency of VHL in promoting proteasomal degradation has led to the recent development of small molecules that hijack the VHL complex to selectively destroy target proteins as a potential therapeutic tool (*Bondeson et al., 2015*). Despite this clear role for proteasomal degradation of HIF, it has been reported that lysosomal inhibitors can lead to stabilisation of the HIFα subunit in both normal oxygen levels and in hypoxia. Moreover, this stabilisation can lead to a functional HIF response (*Lim et al., 2006*), and upregulation of target genes to promote glucose metabolism and angiogenesis (*Hubbi et al., 2013*).

**eLife digest** Most organisms have developed strategies to survive in low oxygen environments. Central to this response are proteins called Hypoxia Inducible Factors (HIFs), which activate genes involved in energy production and blood vessel growth when oxygen is scarce.

When plenty of oxygen is present, HIFs are rapidly broken down. This is important because HIFs have also been linked to the growth and spread of cancers. Oxygen sensing enzymes, termed prolyl hydroxylases, play a principal role in controlling the break down of HIFs when oxygen is abundant. However, the activity of these prolyl hydroxylases can be reduced by changes in the nutrient or iron levels present in the cell. This raises questions about how other cell mechanisms help to control HIF levels.

By using a technique called an unbiased forward genetic screen to study human cells, Miles, Burr et al. set out to identify the cellular pathways that regulate HIF levels when oxygen is still abundant. Disrupting a pump called the V-ATPase – which normally helps to break down unwanted proteins by acidifying the cellular compartments where they are destroyed – stabilised HIFs. Moreover, Miles, Burr et al. identified two previously uncharacterised genes that are required for the V-ATPase to work correctly.

While the V-ATPase is typically associated with the destruction of proteins, a different, unexpected aspect of its activity is responsible for stabilising HIFs. Blocking activity of the V-ATPase reduces levels of iron inside the cell. This inhibits the activity of the prolyl hydroxylases, resulting in HIFs being activated.

Overall, the findings presented by Miles, Burr et al. show key links between oxygen sensing, the use of iron and the V-ATPase. Further work is now needed to investigate how V-ATPase activity affects levels of HIFs found inside cells during diseases such as cancer.

Initial observations regarding lysosomal degradation and HIFs arose from studies using Bafilomycin A (BafA) to chemically inhibit the vacuolar H+ ATPase (V-ATPase), the main complex responsible for acidification of endosomal and lysosomal compartments. BafA treatment stabilised HIF1α and prevented its degradation (*Lim et al., 2006*). Others report similar findings, with several proposed mechanisms to explain the stabilisation of HIF1 upon BafA treatment, including chaperone-mediated autophagy (CMA) (*Bremm et al., 2014*; *Ferreira et al., 2015*; *Hubbi et al., 2014*, *2013*; *Selfridge et al., 2016*), mitochondrial uncoupling (*Zhdanov et al., 2012*) and binding of the V-ATPase to VHL (*Lim et al., 2007*). However, the relative importance of these mechanisms compared to the canonical degradation of HIF1α by prolyl hydroxylation and VHL mediated proteasomal degradation was not clear.

We recently developed a forward genetic screen in near-haploid KBM7 cells to identify genes that regulate HIF1α in aerobic conditions (*Burr et al., 2016*). Here, we used this screen to focus on cellular pathways enriched for gene-trapping insertions, and find that mutations in several V-ATPase subunits result in increased HIF1α levels. In addition, we identify two uncharacterised V-ATPase accessory proteins, TMEM199 and CCDC115, which we show are required for V-ATPase function and form the mammalian orthologue of the yeast Vma12p-Vma22p V-ATPase assembly complex. Although the V-ATPase is required for lysosomal degradation (*Maxson and Grinstein, 2014*), we find that the mechanism for HIF1α stabilisation following V-ATPase inhibition is through intracellular iron depletion, leading to decreased PHD activity. Iron supplementation to V-ATPase depleted cells directly restores PHD hydroxylation of HIF1α in cellular assays and in vitro. These findings support a novel role for the V-ATPase and its assembly factors in regulating HIF1α levels through the control of intracellular iron levels.

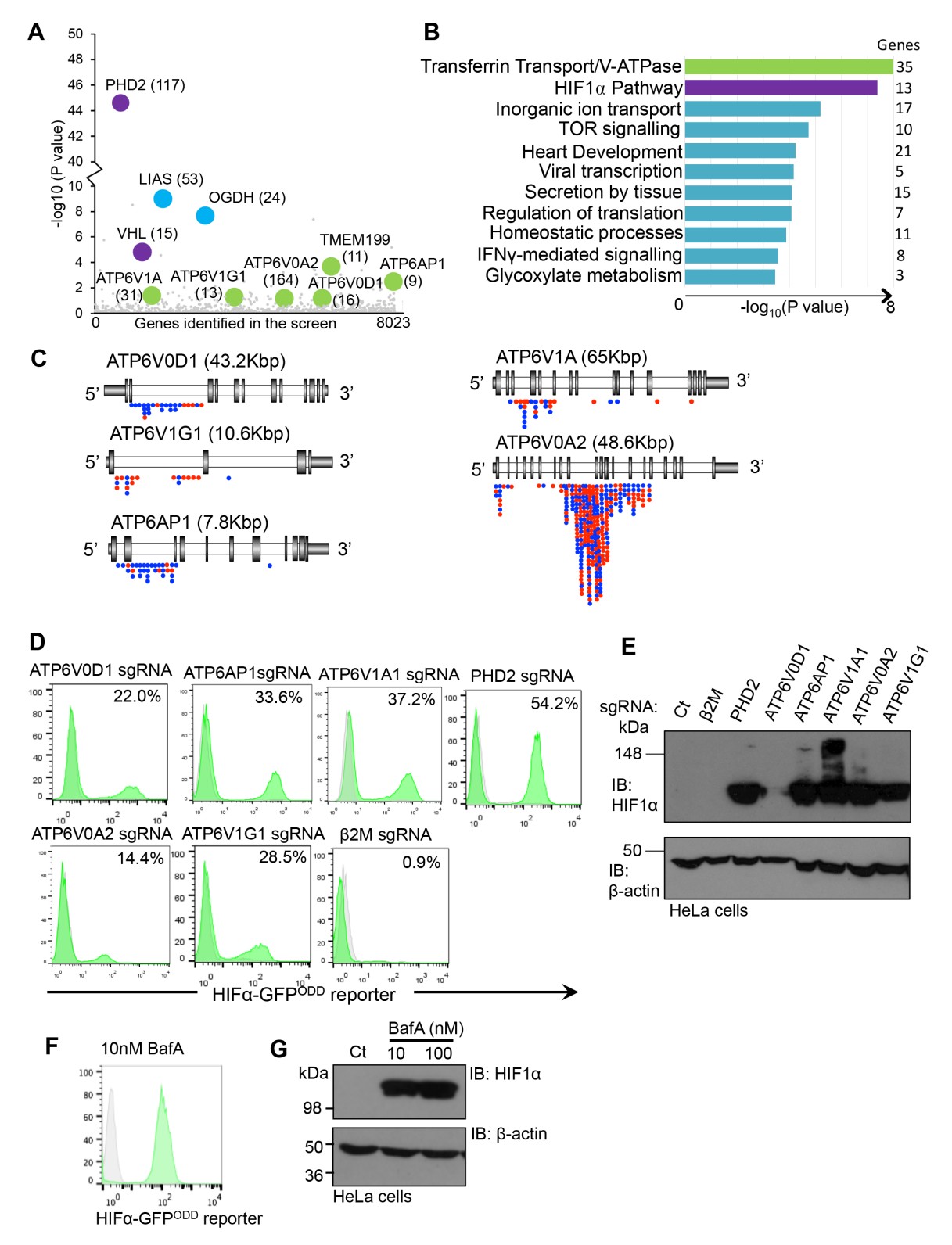

**Figure 1.** Depletion or inhibition of the V-ATPase stabilises HIF1α in aerobic conditions. (**A**) Bubble plot depicting genes enriched in the forward genetic screen. Bubbles represent the genes enriched in the GFP^HIGH population compared to unmutagenised KBM7 cells expressing the HIF1α-GFP^ODD reporter. Proteins involved in V-ATPase assembly and function (green), canonical HIF1α regulation (purple), and the oxoglutarate dehydrogenase complex (blue) are highlighted, with the number of independent gene trap insertions indicated (brackets). (**B**) Pathway analysis of

*Figure 1 continued on next page*

*Figure 1 continued*

enriched genes in the KBM7 forward genetic screen. The top 114 genes enriched for multiple independent gene-trapping integrations in the GFP[HIGH] population compared to unmutagenised KBM7 cells expressing the HIF1α-GFP[ODD] reporter were analysed by gene ontology clustering for pathways significantly targeted in the screen. An individual gene enrichment p value < 0.1 was used as a threshold value for genes to be included in the pathway analysis. (C) Schematic of enriched gene trap insertion sites in the 5 V-ATPase subunits (ATP6V0D1, ATP6V1G1, ATP6AP1, ATP6V1A, ATP6V0A2) identified in the forward genetic screen. (Red = sense insertions, Blue = antisense insertions). (D, E) Validation of the V-ATPase subunits identified in the screen using CRISPR-Cas9 targeted gene editing in HIF1α-GFP[ODD] reporter (D) and wildtype (E) HeLa cells. Cells were simultaneously transduced with Cas9 and sgRNAs to ATP6V0D1, ATP6V1G1, ATP6AP1, ATP6V1A1, or ATP6V0A2. GFP levels were assessed by flow cytometry after 10 days (% GFP[HIGH] cells indicated) (D), and HIF1α levels measured by immunoblot (E). PHD2 and β2m were used as positive and negative controls respectively. (F, G) Chemical perturbation of V-ATPase function. Wildtype and HIF1α-GFP[ODD] HeLa cells were cultured in the presence of BafA (10 nM or 100 nM) for 24 hr and HIF1α levels measured by GFP fluorescence (F) or immunoblot (G).

## Results

### Depletion or inhibition of the V-ATPase stabilises HIF1α in aerobic conditions

We developed a forward genetic screening approach to identify genes involved in the regulation of HIF1α under aerobic conditions using near haploid human KBM7 cells expressing a HIF1α-specific fluorescent reporter (HIF1α-GFP[ODD]) (*Burr et al., 2016*). Briefly, this screen involved randomly muta-genising a clonal population of KBM7 HIF1α-GFP[ODD] reporter cells with a gene-trapping retrovirus, fluorescence activated cell sorting (FACS) to enrich for rare mutations that resulted in increased GFP expression, and mapping the insertion sites in these GFP[HIGH] cells with Illumina HiSeq. This approach successfully identified genes involved in the canonical proteasomal degradation of HIF1α (PHD2 and VHL) as well as genes involved in mitochondrial regulation of HIF1α (oxoglutarate dehydrogenase and lipoic acid synthase) (*Figure 1A*) (*Burr et al., 2016*). In addition, we noted that several other genes were enriched for trapping insertions compared to the control library, although their significance levels were lower (*Figure 1A*). Therefore, we ranked genes most enriched for trapping insertions according to biological process and molecular function (*Figure 1B*, *Supplementary file 1*). As expected, the HIF pathway was ranked highly, due to the enrichment of trapping insertions in genes such as PHD2, VHL and CUL2. However, the top ranked biological process was transferrin transport and V-ATPase function (*Figure 1B*), principally relating to mutagenesis of genes encoding five V-ATPase subunits: ATP6AP1, ATP6V1A, ATP6V1G1, ATP6V0A2 and ATP6V0D1 (*Figure 1A*). Subsequent analysis of the location of V-ATPase gene-trap inserts confirmed that these genes were enriched for mutations in a trapping orientation (*Figure 1C*), consistent with mutations resulting in deletion phenotypes.

We validated whether disrupting the V-ATPase genes stabilised HIF1α using CRISPR (Clustered regularly-interspaced short palindromic repeats)-Cas9 gene-editing in wildtype HeLa cells and those expressing the HIF1α-GFP[ODD] reporter (*Figure 1D,E*). Several single guide RNAs (sgRNA) were designed for each V-ATPase subunit, and transduced along with Cas9 into the HIF1α-GFP[ODD] reporter or wildtype HeLa cells. GFP or endogenous HIF1α levels were measured by flow cytometry or immunoblot at least ten days after transduction. Depletion of all of the V-ATPase subunits identified in the screen increased GFP levels in the sgRNA-targeted cells compared with the wildtype cells and control sgRNA to MHC Class I (β2 microglobulin, β2m) (*Figure 1D*), but to a lesser extent than depletion of the main PHD enzyme for HIF1α, PHD2 (*Figure 1D*). Depletion of the V-ATPase subunits also increased endogenous HIF1α, depending on the efficiency of the sgRNA (*Figure 1E*). Furthermore, chemical inhibition of V-ATPase activity with the inhibitor Bafilomycin A (BafA) activated the GFP reporter, increasing HIF1α levels in aerobic conditions (*Figure 1F,G*) without affecting HIF1α mRNA expression (see Figure 9J).

### TMEM199 and CCDC115 are the human orthologues of the yeast Vma12p-Vma22p V-ATPase assembly complex, and their depletion stabilises HIF1α

In addition to the V-ATPase subunits detected in the screen, we identified that TMEM199 was significantly enriched for gene-trapping insertions (*Figure 2A*). TMEM199 is a putative transmembrane protein with homology (24% sequence identity) to the yeast V-ATPase assembly protein Vma12p

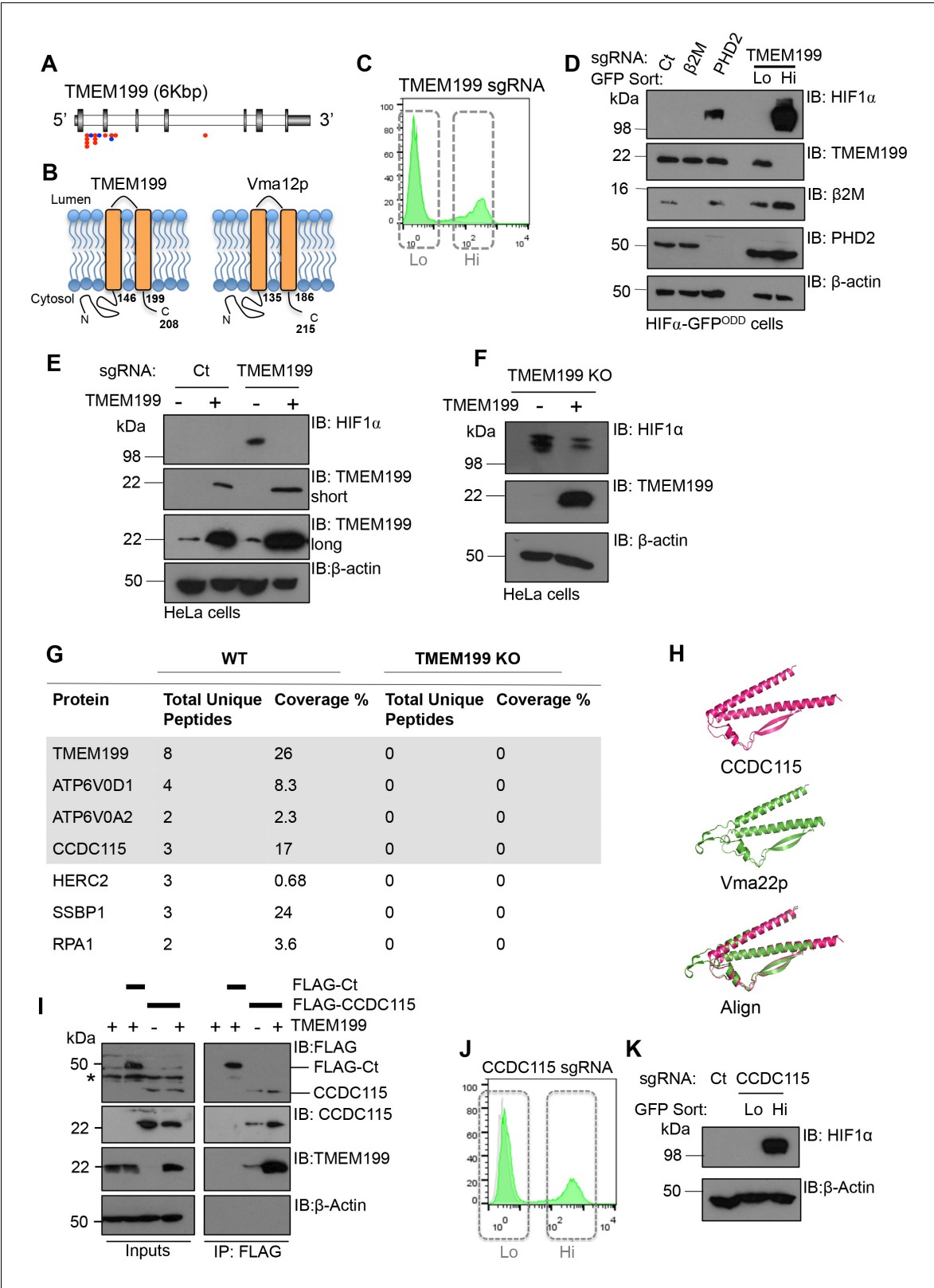

**Figure 2.** TMEM199 and CCDC115 are the human orthologues of the yeast Vma12p-Vma22p V-ATPase assembly complex. (A) Enriched gene trap insertion sites in TMEM199 identified in the forward genetic screen. (Red = sense insertions, Blue = antisense insertions). (B) Schematic for TMEM199 (left) and Vma12p (right) membrane topology. TMEM199 and Vma12p demonstrate 23.89% sequence identity (Clustal Omega tool (EMBL-EBI)). (C, D) HIF1α-GFP[ODD] reporter cells transduced with Cas9/TMEM199 sgRNA were sorted into GFP[LOW] (Lo) and GFP[HIGH] (Hi) populations by FACS (C), lysed,

*Figure 2 continued on next page*

*Figure 2 continued*

and immunoblotted for endogenous HIF1α and TMEM199 (D). PHD2 and β2m were used as positive and negative controls respectively, and β-actin served as a loading control. (E, F) TMEM199 reconstitution decreases HIF1α levels in TMEM199 deficient cells. TMEM199 KO clones were isolated following lentiviral transduction with sgRNA to TMEM199/Cas9 and serial dilution. Null clones were identified by immunoblot. A CRISPR resistant TMEM199 was overexpressed by lentiviral transduction in mixed populations of TMEM199 deficient cells (E) or clonal cells (F). HIF1α and TMEM199 levels were measured by immunoblot, and short and long exposures of TMEM199 levels are shown (E). (G) Co-immunoprecipitation coupled mass spectrometry. Wildtype HeLa cells and TMEM199 null cells were lysed in 1% NP-40 and immunoprecipitated for TMEM199 for 3 hr. Samples were validated by immunoblotting and submitted for mass spectrometry analysis. Proteins immunoprecipitated in wildtype HeLa compared to TMEM199 KO cells with a unique peptide count >2 are shown. (H) PyMOL structural alignment of CCDC115 (pink) and Vma22p (green) based on Phyre$^2$ predictions. (I) Immunoprecipitation of FLAG-CCDC115 with endogenous TMEM199 in wildtype (+) or TMEM199 deficient (-) HeLa cells. An unrelated FLAG tagged protein (FLAG-Ct) was used as a control. The lysate inputs and immunoprecipitated samples are shown. *non-specific band. (J, K) HIF1α-GFP$^{ODD}$ reporter cells were depleted of CCDC115 by transduction with Cas9 and sgRNA. After 12 days, cells were sorted into GFP$^{LOW}$ (Lo, grey box, left) and GFP$^{HIGH}$ (Hi, grey box, right) populations by FACS (J), and immunoblotted for endogenous HIF1α (K). β-actin served as a loading control.

(also known as Vph2p) (*Hirata et al., 1993*) (*Figure 2B*). Depletion of TMEM199 resulted in accumulation of the HIF1α reporter (*Figure 2C*), similarly to levels observed for depletion of the V-ATPase subunits (*Figure 1D*). FACS of the GFP$^{LOW}$ and GFP$^{HIGH}$ TMEM199 sgRNA-targeted cells followed by immunoblot showed that endogenous HIF1α only accumulated in the GFP$^{HIGH}$ population (*Figure 2C,D*). Moreover, when we overexpressed a CRISPR resistant TMEM199 in these TMEM199 sgRNA-targeted cells, the accumulation of HIF1α was reversed (*Figure 2E*). We also isolated TMEM199 knockout (KO) clones, and while most TMEM199 knockouts were lethal after three weeks, a small number of clones that showed undetectable levels of endogenous TMEM199 by immunoblot had elevated HIF1α levels despite several passages (*Figure 2F*). Reconstituting these clonal cells with overexpressed TMEM199 decreased HIF1α levels (*Figure 2F*), further confirming the effect of TMEM199 depletion on HIF1α accumulation.

Human TMEM199 mutations have been recently associated with glycosylation defects (*Jansen et al., 2016c*), but the role of TMEM199 in V-ATPase function was not known. To determine if TMEM199 was involved in the V-ATPase complex, we immunoprecipitated endogenous TMEM199 from wildtype HeLa cells and analysed the associated proteins by mass spectrometry, using the TMEM199 KO HeLa clones as a control (*Figure 2G*). Six proteins were identified as associating with TMEM199 with high confidence compared to the TMEM199 KO cells (*Figure 2G*). Of these, two were V-ATPase subunits, ATP6V0D1 and ATP6V0A2, which were also identified in the genetic screen (*Figure 1A,C*), and have been shown to associate with Vma12p in yeast (*Graham et al., 1998*). A further protein, CCDC115 (coiled-coil domain-containing protein 115), was of particular interest as a 3-dimensional structural prediction analysis (*Kelley et al., 2015*) showed high structural homology to the yeast V-ATPase assembly factor Vma22p (*Figure 2H*) known to bind Vma12p (*Graham et al., 1998*), and human mutations in CCDC115 have been recently reported to show glycosylation defects (*Jansen et al., 2016b*) similarly to the TMEM199 mutations. Immunoprecipitation of endogenous TMEM199 with FLAG-tagged CCDC115 confirmed the interaction identified by mass spectrometry (*Figure 2I*), and depleting CCDC115 from HeLa HIF1α-GFP$^{ODD}$ cells stabilised the fluorescent reporter and endogenous HIF1α similarly to TMEM199 deletion (*Figure 2J,K*). Thus, it was likely that TMEM199 and CCDC115 were mammalian homologues of the yeast V-ATPase assembly proteins.

## TMEM199 and CCDC115 localise to the ER and are required for endolysosomal acidification and lysosomal degradation

Yeast Vma12p-Vma22p localise to the endoplasmic reticulum (ER), where they are thought to be involved in V-ATPase assembly (*Graham et al., 1998*). Using cell fractionation experiments we observed that TMEM199 was only present in the total membrane pool and not in the soluble fraction (*Figure 3A*). Furthermore, immunofluorescence microscopy showed endogenous TMEM199 localised predominantly to the ER, rather than endosomal or lysosomal compartments (*Figure 3B,C*). While it was not possible to visualise endogenous CCDC115 by immunoblot or fluorescence microscopy, we observed co-localisation of endogenous TMEM199 with HA-CCDC115 at the ER, although CCDC115 is mostly cytosolic (*Figure 3D,E*). These results suggest that TMEM199 and CCDC115

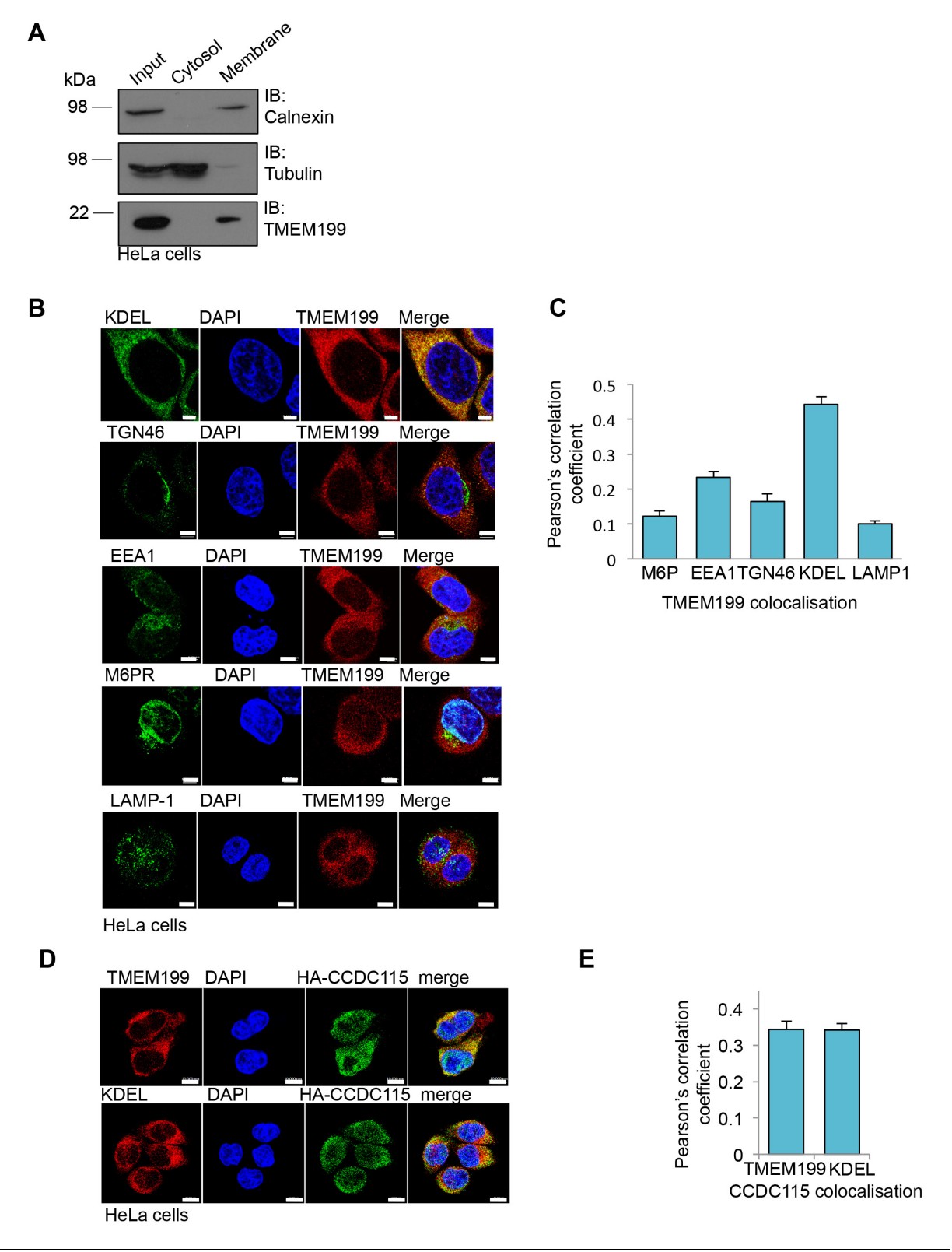

**Figure 3.** TMEM199 and CCDC115 localise to the ER. (**A**) HeLa cells were homogenised and separated into membrane and cytosolic fractions by ultra-centrifugation. The samples were analysed by immunoblotting for TMEM199. Calnexin was used as a loading control for membrane compartments, whilst tubulin was used as a control for cytosolic fractions. (**B, C**) Representative immunocytochemical staining for endogenous TMEM199 (red) with the ER marker KDEL, the golgi apparatus marker TGN46, early endosome marker EEA1, late endosome marker M6PR and lysosomal marker LAMP-1 (all in

*Figure 3 continued on next page*

*Figure 3 continued*

green) (B). Scale bar represents 5 µm. Quantification of colocalisation for TMEM199 and the respective organelle markers using Pearson's Correlation Coefficient (C) n ≥ 16 cells. (D, E) Confocal immunofluorescence microscopy of HeLa cells transduced with HA-CCDC115 (green) and endogenous TMEM199 (red, top) or KDEL (red, bottom) (D). Scale bar represents 10 µm. Quantification of colocalisation for CCDC115 with TMEM199 or KDEL using Pearson's Correlation Coefficient (E) n ≥ 50 cells.

form a complex at the ER, analogous to the yeast Vma12p-Vma22p V-ATPase assembly proteins (*Graham et al., 1998*).

As a major function of the V-ATPase is facilitating the lysosomal degradation of proteins by acidification of endosomal-lysosomal compartments, we examined whether TMEM199 or CCDC115 were required for the degradation of a known lysosomal substrate, epidermal growth factor receptor (EGFR). We sorted for TMEM199 or CCDC115 KOs in HIF1α-GFP$^{ODD}$ reporter Hela cells by FACS, and measured EGFR degradation following EGF stimulation using BafA as a control (*Figure 4A–C*). EGF treatment stimulated the degradation of EGFR at 45 min in the control cells, with near complete loss of the receptor by 3 hr, consistent with prior reports (*Mizuno et al., 2005*) (*Figure 4A–C*). This degradation was prevented with BafA treatment (*Figure 4A*), confirming the role of the V-ATPase in the degradation of EGFR. However, EGF had no effect on the levels of EGFR in the TMEM199 or CCDC15 null cells, and no degradation was detected over 3 hr (*Figure 4B,C*). Indeed, TMEM199 and CCDC115 deficient cells prevented EGFR degradation similarly to BafA treatment (*Figure 4A*).

The requirement for TMEM199 and CCDC115 in lysosomal degradation was not limited to EGFR, as we observed impaired degradation of another lysosomal substrate, Major Histocompatibility Complex (MHC) Class I. The degradation of cell surface MHC Class I molecules by the Kaposi Sarcoma Herpesvirus (KSHV) gene product K3 is a well characterised pathway dependent on ubiquitination and lysosomal degradation (*Coscoy and Ganem, 2000*; *Hewitt et al., 2002*; *Ishido et al., 2000*). Therefore, we used HeLa cells expressing K3, which have low resting levels of cell surface MHC Class I (*Figure 4D*), and transduced these cells with sgRNA targeting TMEM199, CCDC115 or core V-ATPase subunits (ATP6V1A1, ATP6V0D1). Mixed populations of TMEM199 or CCDC115 deficient cells partially rescued MHC Class I at the cell surface (*Figure 4D*). Similar findings were observed with depletion of the core V-ATPase subunits, consistent with a role for TMEM199 and CCDC115 in V-ATPase facilitated lysosomal degradation (*Figure 4D*).

As the V-ATPase is also required to acidify endosomal compartments, we explored the role of TMEM199 and CCDC115 in endosomal acidification using a fluorescent pH sensitive transferrin receptor (Tfnr-phl) (*Merrifield et al., 2005*). This construct encodes a super-ecliptic GFP phlourin attached to the extracellular domain of the receptor, which is quenched on transition from pH 7 to 5 (*Merrifield et al., 2005*) (*Figure 5—figure supplement 1A*), and therefore not visible in acidified compartments. To examine how endosomal pH (e.g. Tfnr-phl fluorescence) was associated with HIF1α stabilisation, we substituted GFP in our HIF1α reporter construct to generate HIF1α-mCherry$^{ODD}$ reporter cells. Live cell microscopy of these cells expressing Tfnr-phl showed GFP fluorescence only at the cell surface (*Figure 5A,B*), as the transferrin receptor typically recycles between the plasma membrane and recycling vesicles (*Maxfield and McGraw, 2004*), and is quenched in the acidic endosomal compartments. Confocal fluorescence microscopy of fixed Tfnr-phl expressing cells (i.e. no longer pH sensitive) confirmed that the receptor was still present in endosomal compartments (*Figure 5—figure supplement 1B*). However, live cell microscopy of HIF1α-mCherry$^{ODD}$ reporter cells treated with BafA revealed Tfnr-phl within intracellular vesicles, confirming that V-ATPase inhibition prevented acidification of endosomal compartments (*Figure 5A,B*). Indeed, the localisation of Tfnr-phl following BafA treatment was similar to the confocal fluorescence microscopy of fixed Tfnr-phl expressing cells (*Figure 5—figure supplement 1B*). Moreover, mCherry fluorescence was only observed in the BafA treated cells (*Figure 5A*), confirming that V-ATPase inhibition led to decreased endosomal acidification and HIF1α stabilisation.

We next examined the effect of TMEM199 or CCDC115 depletion on Tfnr-phl intracellular fluorescence in HIF1α-mCherry$^{ODD}$ HeLa reporter cells (*Figure 5C*), using mCherry accumulation to identify cells where V-ATPase formation was disrupted. Mixed KO populations of the core V-ATPase subunits ATP6V1A1 or ATP6V0D1 were used as a control. Live cell imaging of transiently transfected

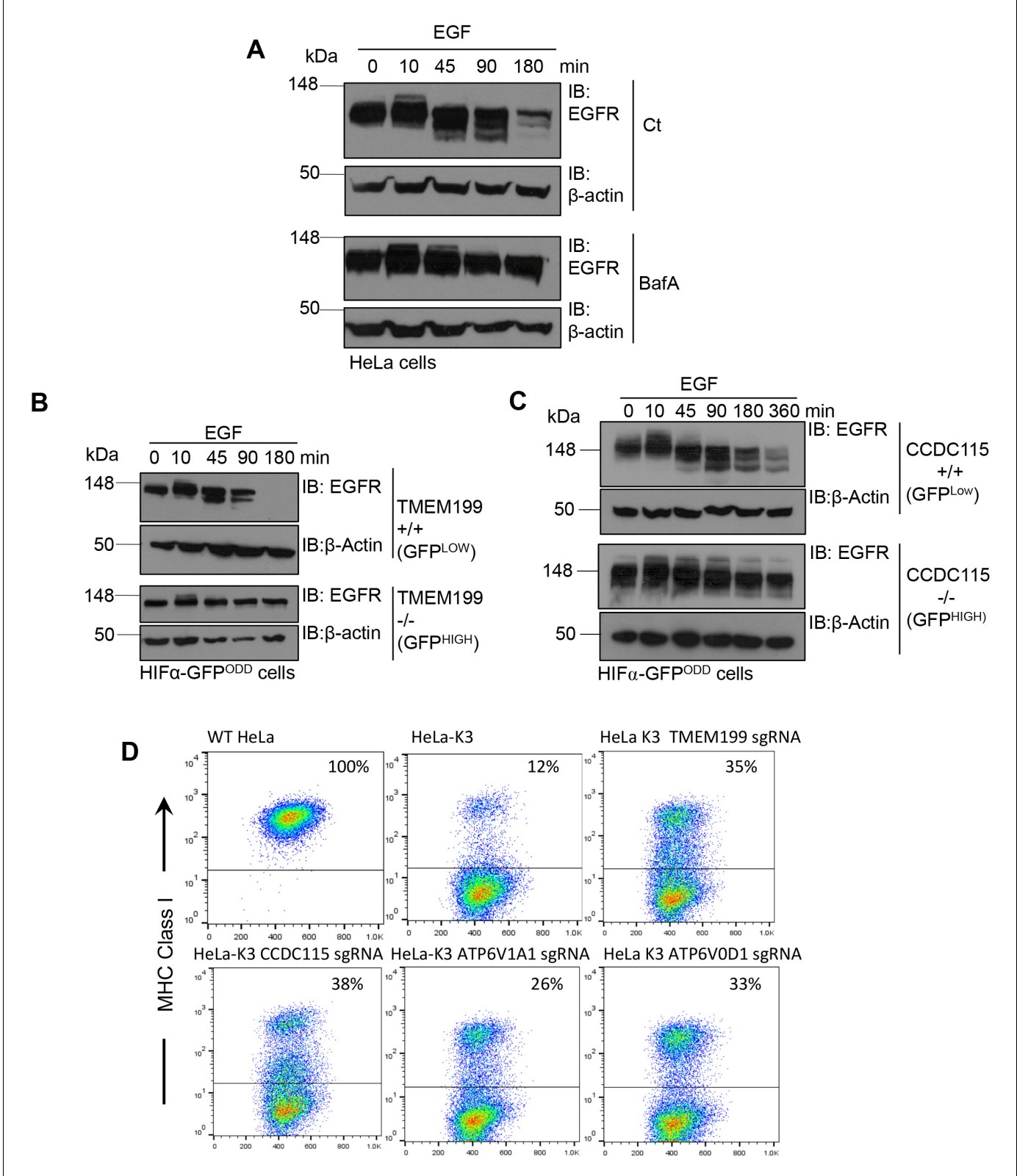

**Figure 4.** TMEM199 and CCDC115 and are required for lysosomal degradation of EGFR and MHC Class I. (**A**) EGFR degradation assay for wildtype and BafA treated cells. HeLa cells were cultured in the presence or absence of 10 nM BafA for 24 hr. Cells were stimulated with EGF and lysed at the indicated times. Lysates were subjected to SDS-PAGE and immunoblotted for EGFR. β-actin was used as a loading control. (**B, C**) EGFR degradation assay for TMEM119 and CCDC115 deficient cells. HIF1α-GFP^ODD cells were transduced with Cas9 and sgRNA to TMEM199 (**B**) or CCDC115 (**C**). After

*Figure 4 continued on next page*

Figure 4 continued

14 days, cells were sorted into TMEM199 or CCDC115 sufficient (+/+, GFP$^{LOW}$), and TMEM199 or CCDC115 null (−/−, GFP$^{HIGH}$) populations as described. Cells were then cultured for 24 hr before stimulation with EGF (100 ng/ml), harvested at indicated times and immunoblotted for EGFR. (**D**) MHC Class I degradation in HeLa cells expressing K3. HeLa-K3 cells were transduced with Cas9 and sgRNA to TMEM199, CCDC115, ATP6V1A1 or ATP6V0D1. After 14 days, cell surface MHC Class I levels were measured by flow cytometry (mAb W6/32). Wildtype HeLa cells were used as a control for total MHC Class I. Percentages of cells with MHC Class I at the cell surface are shown.

Tfnr-phl in mixed KO populations of TMEM199, CCDC115, or the core V-ATPase subunits showed that Tfnr-phl intracellular fluorescence was almost entirely observed in cells that stabilised the HIF1α-mCherry$^{ODD}$ reporter (*Figure 5C,D* and *Figure 5—figure supplement 1C*). Thus, TMEM199 and CCDC115 depletion prevents acidification of endosomes, similarly to BafA treatment or depletion of core V-ATPase subunits.

## V-ATPase depletion activates HIF1 by preventing HIF1α prolyl hydroxylation

The identification of V-ATPase components as regulators of HIF1α levels by a genetic approach was unexpected, given that depletion of PHD2 and VHL, the principal genes involved in the canonical pathway for HIF1α proteasomal degradation, are sufficient for HIF1 activation. We therefore measured the levels of HIF1α in HeLa cells treated with different V-ATPase inhibitors in comparison to the proteasome inhibitor, MG132 (*Figure 6A*). Surprisingly, we observed similar levels of HIF1α accumulation following treatment with either V-ATPase inhibitors or MG132. Furthermore, BafA treatment increased HIF1α levels after just 4 hr (*Figure 6B*). Rather than observing HIF1α accumulation within endosomal-lysosomal compartments, as would be expected following disruption of the V-ATPase, HIF1α was only observed in the nuclei of cells treated with BafA (*Figure 6C*) or depleted of TMEM199 (*Figure 6D*). This nuclear increase in HIF1α was sufficient to activate HIF responsive genes, demonstrated by an increase in cell surface and total Carbonic Anhydrase 9 (CA9) when cells were depleted of TMEM199, CCDC115 or core V-ATPase subunits (*Figure 6E,F*). V-ATPase inhibition or depletion also stabilised HIF2α in HeLa cells and the HIF target gene Heme oxygenase 1 (HO-1) (*Bertout et al., 2009*) (*Figure 6G*).

Prior studies suggest that CMA may contribute to the degradation of HIF1α (*Hubbi et al., 2013*). To determine if CMA or macroautophagy played a significant role in the regulation of HIF1α in aerobic conditions, we examined the effect of depletion of HSC70, LAMP2A and ATG16 on HIF1α levels. Depletion of the CMA mediators, HSC70 and LAMP2A, or loss of the macroautophagy protein, ATG16, had no effect on HIF1α stabilisation in aerobic conditions (*Figure 7A–C*). Moreover, BafA treatment still stabilised HIF1α even when LAMP2A, HSC70 or ATG16 were depleted (*Figure 7A,C, D*). We therefore sought alternative explanations to account for the accumulation of HIF1α following V-ATPase inhibition, and focused on the principle mechanism for regulating HIF1α stability, prolyl-hydroxylation.

We measured the levels of total and prolyl-hydroxylated HIF1α in HeLa cells treated with several V-ATPase inhibitors, the proteasome inhibitor MG132, or the PHD inhibitor DMOG (dimethyloxaloyl-glycine) using a HIF1α prolyl hydroxyl-specific antibody (*Figure 7E,F*). While all inhibitors increased total HIF1α levels, only MG132 resulted in the accumulation of prolyl-hydroxylated HIF1α (*Figure 7E,F*). Conversely, BafA and chloroquine stabilised HIF1α in a non-hydroxylated form, similarly to DMOG treatment (*Figure 7E,F*).

To directly measure if PHD activity was reduced following V-ATPase inhibition, we used an in vitro assay of HIF1α prolyl-hydroxylation (*Burr et al., 2016*), which allows measurements for PHD activity from cell lysates without the addition of excess cofactors (*Figure 7G,H*). Lysates from wildtype HeLa cells or cells treated with BafA for 24 hr were incubated with a purified His-tagged HIF1α$^{ODD}$ protein for 15 min, and hydroxylation measured using the hydroxyprolyl-specific antibody. While HIF1α$^{ODD}$ was rapidly hydroxylated in the wildtype HeLa lysate, BafA treatment markedly reduced hydroxylation (*Figure 7G,H*). Thus, rather than preventing the lysosomal degradation of HIF1α, the V-ATPase inhibition stabilised HIF1α by decreasing PHD enzymatic activity.

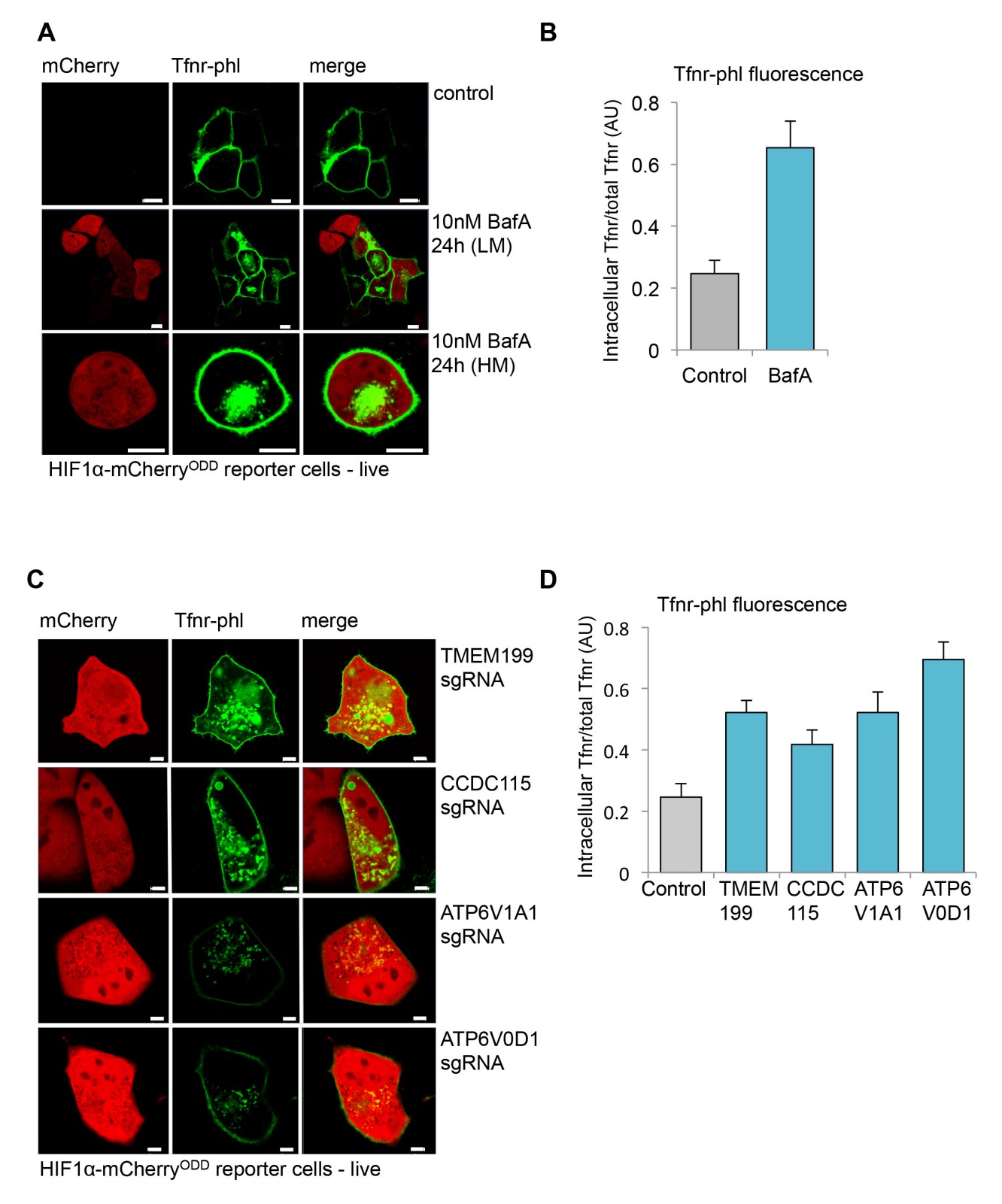

**Figure 5.** TMEM199 and CCDC115 are required for acidification of endosomal compartments. (**A, B**) Live cell confocal microscopy of HIF1α-mCherry$^{ODD}$ reporter cells transfected with the pH sensitive Tfnr-phl. HIF1α-mCherry$^{ODD}$ reporter cells were transfected with Tfnr-phl and treated with or without 10 nM BafA for 24 hr. Lower (LM) and Higher (HM) magnifications of representative BafA treated cells are shown (**A**). Quantification of intracellular Tfnr-phl/total Tfnr-phl fluorescence in BafA treated cells compared to no treatment (≥9 cells) (**B**). Additional control experiments are shown

*Figure 5 continued on next page*

*Figure 5 continued*

in *Figure 5—figure supplement 1*. (C, D) Live cell confocal microscopy of Tnfr-phl fluorescence in HIF1α-mCherry<sup>ODD</sup> reporter cells depleted for TMEM199, CCDC115 or core V-ATPase subunits. TMEM199, CCDC115, ATP6V1A1 or ATP6V0D1 were depleted by sgRNA as described. After 10–12 days the cells were transfected with Tfnr-phl and live cell fluorescence measured after a further 24 to 48 hr (C). Quantification of intracellular Tfnr-phl/ total Tfnr-phl fluorescence is shown (≥12 cells) (D). Representative wide field images are shown in *Figure 5—figure supplement 1C*. Scale bars represent 10 μm (A) or 5 μm (C). Values are mean±SEM.

The following figure supplement is available for figure 5:

**Figure supplement 1.** TMEM199 and CCDC115 are required for acidification of endosomal compartments.

## Disrupting the V-ATPase decreases intracellular iron levels and prevents HIF1α prolyl hydroxylation

PHDs are 2-oxoglutarate (2-OG) dependent dioxygenases, which aside from molecular oxygen, require Fe(II) and 2-OG as cofactors for catalytic activity. As hydroxylation of HIF1α was impaired by V-ATPase inhibition in aerobic conditions, it was possible that V-ATPase activity altered the level of PHD cofactors. We focused on iron as: (i) iron chelators and iron metabolism can alter PHD activity (*Wang and Semenza, 1993*), and (ii) the V-ATPase is implicated in iron homeostasis via clathrin-mediated endocytosis of transferrin (*Kozik et al., 2013*), the conversion of ferric to ferrous iron within endosomes (*Dautry-Varsat et al., 1983*; *Straud et al., 2010*), and the release of iron from ferritin stores (*Mancias et al., 2014*).

To examine if V-ATPase inhibition resulted in cytosolic iron depletion, we treated the cells with BafA and measured the levels of IRP2 (also known as IREB2); a sensitive maker of intracellular free iron that is rapidly ubiquitinated and undergoes proteasomal degradation in iron-replete cells, but accumulates in iron deficient conditions (*Iwai et al., 1995*; *Salahudeen et al., 2009*; *Vashisht et al., 2009*). The iron chelator, desferrioxamine (DFO), which has an established role in inhibiting PHD activity and activating the hypoxia pathway, was used as a control for intracellular iron depletion, (*Jaakkola et al., 2001*; *Wang and Semenza, 1993*). BafA treatment stabilised IRP2 similarly to cells treated with DFO (*Figure 8A*), consistent with V-ATPase inhibition leading to depletion of intracellular iron. DFO and particularly BafA treatment also increased NCOA4, the main cargo receptor for the autophagic degradation of ferritin (ferritinophagy) and mobilisation of intracellular iron stores (*Mancias et al., 2014*). However, while NCOA4 promoted ferritinophagy in DFO treated cells, ferritin levels were unchanged in BafA treated cells, consistent with a complete block in autophagy when the V-ATPase was inhibited (*Figure 8A*). Similar increases in IRP2 and NCOA4 were observed in FACS GFP<sup>HIGH</sup> populations of TMEM199, CCDC115, ATP6V0D1 or ATP6V1A1 deficient HIF1α-GFP<sup>ODD</sup> reporter cells (*Figure 8B*). Together, these findings demonstrate that disrupting V-ATPase integrity and assembly results in intracellular iron depletion.

We next asked if replenishing ferrous iron levels directly restored HIF1α prolyl hydroxylation. We first validated that ferrous iron restored in vitro prolyl hydroxylation of the HIF1α<sup>ODD</sup> in DFO treated lysates (*Figure 8C,D*). We then examined the effect of ferrous iron treatment on PHD activity in extracts from BafA treated HeLa cells (*Figure 8E,F*). The addition of 1 μM Fe(II) completely restored prolyl hydroxylation of the HIF1α<sup>ODD</sup> (*Figure 8E,F*). Indeed, the levels of HIF1α hydroxylation were slightly higher than observed in wildtype cells, although this increase was not significant.

## Iron supplementation restores HIF1α turnover in cells when V-ATPase activity is disrupted

To explore if iron supplementation was sufficient to restore HIF1α turnover in cells where the V-ATPase had been inhibited, BafA treated HIF1α-GFP<sup>ODD</sup> reporter or wildtype HeLa cells were supplemented with or without 50 μM iron (Fe(III)) citrate for 24 hr, and HIF1α levels measured by flow cytometry and immuoblot (*Figure 9A,B*). While BafA treatment stabilised the GFP reporter and endogenous HIF1α, this was completely prevented by the addition of iron to the media. Similar findings were observed in cells treated with DFO and iron citrate (*Figure 9C,D*), although the concentration of iron needed to restore HIF1α degradation was higher (200 μM Fe(III)). Iron supplementation also restored HIF1α turnover in BafA treated primary human dermal fibroblasts, HEK293ET cells and RCC10 cells that had been reconstituted with VHL (*Figure 9E–G*). Furthermore, iron citrate also

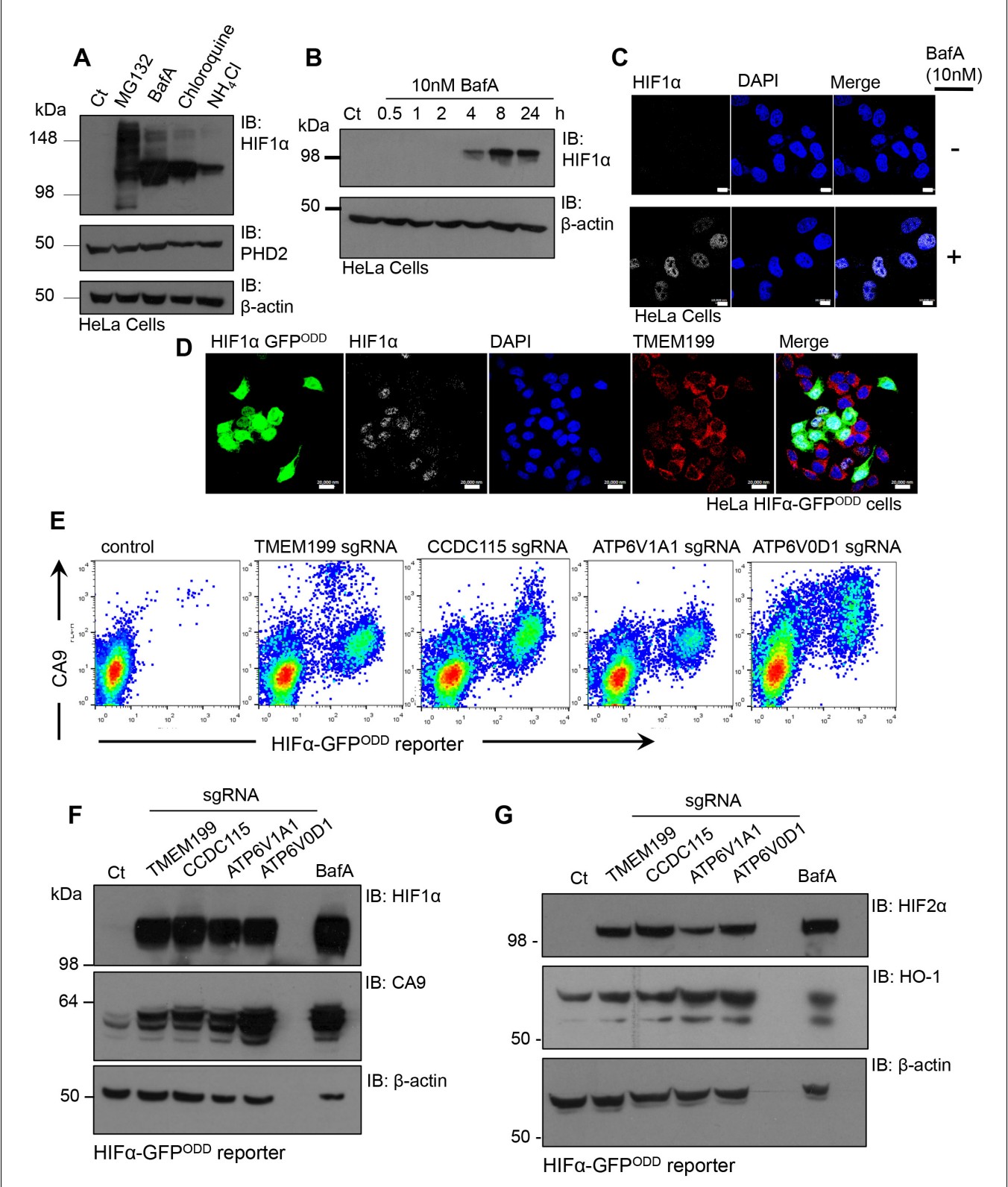

**Figure 6.** Disrupting the V-ATPase activates HIF1 and HIF2. (**A**) Immunoblot of HIF1α levels in response to the proteasome inhibitor MG132, the V-ATPase inhibitor BafA, the lysosomotropic agent Chloroquine, and the oxidative metabolism inhibitor NH₄Cl. (**B**) Immunoblot of HIF1α levels in HeLa cells in response to BafA treatment at 0.5, 1, 2, 4, 8 and 24 hr. (**C**) Confocal immunofluorescence microscopy of WT (top) and BafA (bottom) treated HeLa cells stained for endogenous HIF1α. Cells were treated in the presence or absence of 10 nM BafA for 24 hr before immunofluorescence staining

*Figure 6 continued on next page*

*Figure 6 continued*

with HIF1α (white). Cells were mounted using DAPI (blue) and visualised by confocal microscopy. Scale bar represents 10 µm. (D) Immunocytochemical staining to examine HIF1α stabilisation in TMEM199 depleted HIF1α-GFP^ODD reporter cells. HIF1α-GFP^ODD reporter cells were depleted of TMEM199 using CRISPR-Cas9 genetics and stained for HIF1α (white), TMEM199 (red) and DAPI (blue). Scale bar represents 20 µm. (E–G) Levels of HIF1α or HIF2α and their target genes in cells depleted of V-ATPase subunits. HIF1α-GFP^ODD reporter cells were transduced with sgRNA to the indicated V-ATPase subunits as described. After 14 days, cell surface CA9 was measured by flow cytometry (E). Levels of HIF1α, HIF2α and their targets CA9 and HO-1 were measured by immunoblot (F, G).

prevented the downstream activation of several HIF1 target genes (GLUT1, VEGF and carbonic anhydrase 9) following BafA treatment without affecting HIF1α mRNA levels (*Figure 9J*). Consistent with iron supplementation only affecting HIF1α levels following V-ATPase inhibition, iron citrate did not affect HIF1α stabilisation following proteasome inhibition or in VHL null RCC10 cells (*Figure 9G, H*). Iron citrate had no direct effect on lysosomal degradation, as EGFR turnover was still impaired in BafA treated cells following iron supplementation (*Figure 9I*).

Although iron citrate restored HIF1α to basal levels following BafA treatment, it was unclear whether HIF1α stabilisation following genetic disruption of the V-ATPase could be prevented by iron supplementation. Therefore, iron citrate was supplemented to the media of mixed KO populations of individual V-ATPase subunits (TMEM199, CCDC115 or ATP6V1A1) and HIF1α levels measured by immunoblot (*Figure 9K–M*). Iron treatment reversed the HIF1α stabilisation in the sgRNA-targeted cells (*Figure 9K–M*), confirming that iron supplementation was sufficient to restore HIF1α degradation following genetic disruption of the V-ATPase.

While it was evident that V-ATPase inhibition resulted in iron depletion, we wanted to explore the relative importance of disrupting iron uptake compared to mobilisation of iron stores. Moreover, the contribution of transferrin independent iron uptake to the intracellular pool was unclear (*Liuzzi et al., 2006*; *Oudit et al., 2006*). We examined if treating cells with ferric iron prior to BafA treatment prevented HIF1α accumulation and conversely, also measured if iron supplementation after BafA treatment affected HIF1α levels (*Figure 10A*). Pre-conditioning the media with iron citrate did not prevent the accumulation of HIF1α following BafA treatment (*Figure 10A*). However, iron supplementation at the time of BafA treatment or after 12 hr did decrease HIF1α levels (*Figure 10A*), suggesting that iron supplementation restores HIF1α turnover following disruption of the V-ATPase by a transferrin independent mechanism.

To further explore the relative contributions of iron uptake and ferritin turnover to intracellular iron homeostasis, we depleted cells of transferrin receptor, IRP2 or NCOA4 by sgRNA, and used HIF1α stability as a sensitive functional measure of iron levels. Mixed KO populations of NCOA4 cells had no effect on HIF1α levels (*Figure 10B,C* – *Figure 10—figure supplement 1A*). Interestingly, NCOA4 depletion did increase IRP2 levels after 12 days, suggesting that IRP2 may increase iron flux through the endosomal pathway, but cell surface transferrin receptor and total ferritin levels did not change (*Figure 10B,C*) Depletion of either IPR2 or the transferrin receptor stabilised the HIF1α-GFP^ODD reporter and endogenous HIF1α, although this was most marked in the transferrin receptor null cells (*Figure 10D–F,H* – *Figure 10—figure supplement 1B–D*). IRP2 depletion also did not alter cell surface transferrin receptor levels (*Figure 10D*), consistent with IRP2 regulating iron flux rather than just transferrin receptor expression. Thus, ferritinophagy does not seem to contribute significantly to cytosolic Fe(II) levels when the transferrin pathway is functional.

Lastly, we examined if iron supplementation restored HIF1α turnover in transferrin receptor deficient cells. Mixed transferrin receptor KO populations were treated with iron citrate for 24 hr and HIF1α levels measured. Iron treatment decreased HIF1α levels and increased ferritin stores without altering NCOA4 levels (*Figure 10G,H* – *Figure 10—figure supplement 1E*). Thus, when the transferrin pathway is impaired, either by depletion of the transferrin receptor or disrupting V-ATPase activity, increasing the availability of extracellular Fe(III) is sufficient to restore PHD activity and HIF1α degradation.

## Discussion

The use of a forward genetic approach to examine cellular processes that regulate HIF1α revealed that disruption of the V-ATPase complex stabilised HIF1α in aerobic conditions. HIF1α accumulation

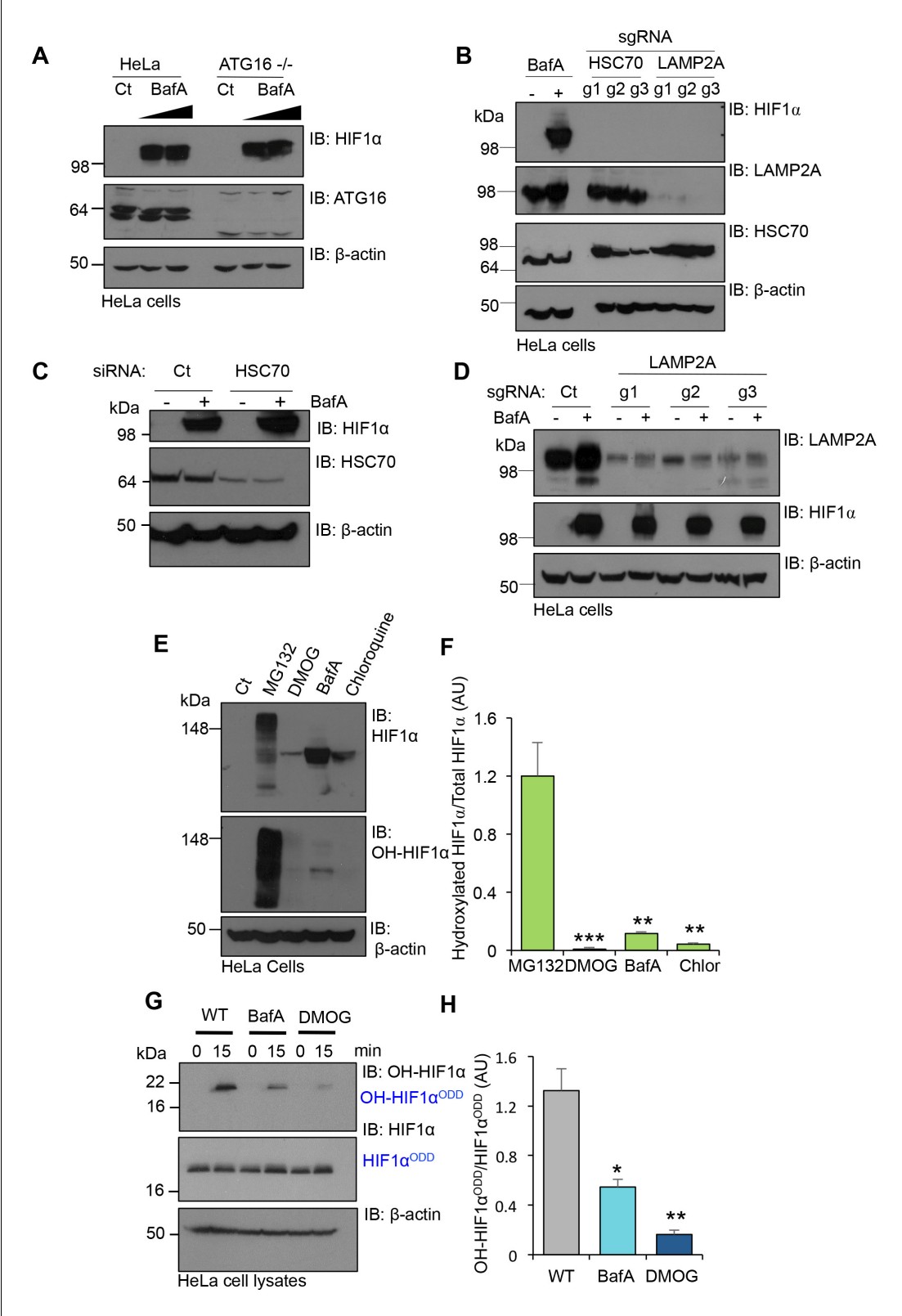

**Figure 7.** V-ATPase depletion or inhibition stabilises HIF1α in a non-prolyl hydroxylated form. (**A**) HIF1α stabilisation in ATG16 null HeLa cells. HeLa cells and ATG16 null cells were treated with increasing concentrations of BafA (10 nM and 100 nM) before immunoblotting for HIF1α. (**B**) HIF1α levels following depletion of HSC70 and LAMP2A in aerobic conditions. HSC70 and LAMP2A depleted cells were generated using CRISPR-Cas9 gene editing with three individual sgRNAs (g1, g2, g3). HIF1α, LAMP2A and HSC70 levels were visualised by immunoblot. Untreated (Ct) and BafA treated HeLa cells

*Figure 7 continued on next page*

*Figure 7 continued*

were used as controls. (**C**) HIF1α levels following siRNA-mediated depletion of HSC70. HeLa cells were transfected with siRNA to HSC70 or an siRNA control (Ct), and HIF1α or HSC70 levels measured by immunoblot after 96 hr. Cells were treated with or without 10 nM BafA for 24 hr prior to lysis. (**D**) LAMP2A deficient HeLa cells were treated with or without 10 nM BafA for 24 hr. Three different sgRNAs were used (g1, g2, g3). (**E, F**) Immunoblot of total HIF1α and the prolyl hydroxylated form in response to MG132, DMOG, BafA and Chloroquine (**E**). Quantification of immunoblots represented using ImageJ analysis (**F**) (n = 3). (**G, H**) In vitro prolyl hydroxylation of the HIF1α$^{ODD}$ protein following incubation with lysates from WT, BafA and DMOG treated HeLa cells. The levels of hydroxylated HIF1α were measured using a prolyl hydroxy-HIF1α specific antibody (**G**). Quantification of the in vitro hydroxylation assay using ImageJ analysis (**H**) (n = 3). Values are mean±SEM. *p<0.05, **p<0.01, ***p<0.001.

following inhibition of V-ATPase activity was unexpected, given the clear role for the proteasome in degrading ubiquitinated HIF1α in normal oxygen tensions. Rather than V-ATPase inhibition directly preventing HIF1α degradation, we find that HIF1α is stabilised due to depletion of intracellular iron, which results from decreased transferrin uptake and reduced conversion to ferrous iron within endo-somal compartments. These findings may have diverse physiological implications, particularly as germline mutations in V-ATPase subunits are associated with several human diseases. Homozygous mutations in a V0 subunit isoform (ATP6V0A3) cause infantile osteopetrosis (failure of bone resorption) (*Kornak et al., 2000*), while mutations in kidney specific isoforms (ATP6V0A4) cause distal tubular renal acidosis (*Karet et al., 1999*; *Smith et al., 2000*). Furthermore, several cancer cell types express V-ATPase complexes at the plasma membrane (*Capecci and Forgac, 2013*), possibly as a mechanism for regulating cytosolic pH, which appear to render tumour cells more susceptible to cell death when the V-ATPase is inhibited (*Perut et al., 2014*). Given the diverse role of the V-ATPase, it will be of future interest to examine whether known germline or somatic human mutations in V-ATPase subunits impact on the HIF pathway in animal models.

While HIF1α degradation by proteasome independent mechanisms, such as CMA, has been reported (*Bremm et al., 2014*; *Ferreira et al., 2013*; *Hubbi et al., 2014*, *2013*; *Selfridge et al., 2016*), we did not observe any effect on HIF1α levels in HeLa cells when key mediators of CMA were depleted (HSC70 and LAMP2A). The ability of iron treatment to completely restore HIF1α turn-over when the V-ATPase is inhibited, without altering the lysosomal degradation of EGFR or the autophagy cargo receptor NCOA4, also argues against a role for CMA in HIF1α regulation. It remains possible that non-proteasomal degradation of HIF1α may occur in certain cell types and under particular conditions. The requirement for the AAA ATPase P97 in HIF1α degradation suggests that it may be incorporated into larger complexes that require unfolding prior to proteolysis (*Alexandru et al., 2008*). Nevertheless, we find that the major consequence of V-ATPase inhibition is stabilisation of HIF1α by preventing prolyl hydroxylation.

Our findings are consistent with prior studies showing impaired uptake of transferrin following V-ATPase inhibition (*Kozik et al., 2013*; *Straud et al., 2010*), and confirm the importance of V-ATPase activity in regulating iron through clathrin-mediated endocytosis (*Kozik et al., 2013*), endosomal acidification (*Eckenroth et al., 2011*; *Ohgami et al., 2005*) and ferritinophagy (*Mancias et al., 2014*). Directly measuring the levels of free intracellular iron within the cytosol is challenging, as reagents typically used for these assays rely on indirect enzymatic assays, and most Fe(II) is rapidly bound to enzymes or incorporated into the biosynthesis of iron-sulphur complexes. However, IRP2 is a sensitive marker of intracellular iron availability (*Iwai et al., 1995*; *Salahudeen et al., 2009*; *Vashisht et al., 2009*), and its induction following V-ATPase inhibition is consistent with depletion of ferrous iron. Furthermore, our in vitro hydroxylation assay clearly shows that supplementation with Fe(II) can completely restore prolyl hydroxylation of HIF1α, confirming that V-ATPase inhibition reduces the available pool of intracellular Fe(II).

The V-ATPase is not only implicated in iron uptake and conversion to Fe(II), but also required for the release of iron from ferritin stores via ferritinophagy. Using HIF1α stabilisation as a sensitive marker for intracellular iron, we find that inhibition or disruption of the V-ATPase both alters iron uptake and degradation of ferritin. However, HIF1α was only stabilised in the transferrin receptor deficient cells, implying that increased ferritin turnover is not sufficient to compensate for prolonged loss of transferrin-mediated iron uptake. Conversely, while loss of NCOA4 activated IRP2, there was still sufficient intracellular iron for PHDs to function and HIF1α was not stabilised. These findings are consistent with decreased transferrin uptake and conversion of Fe(III) to Fe(II) being the predominant

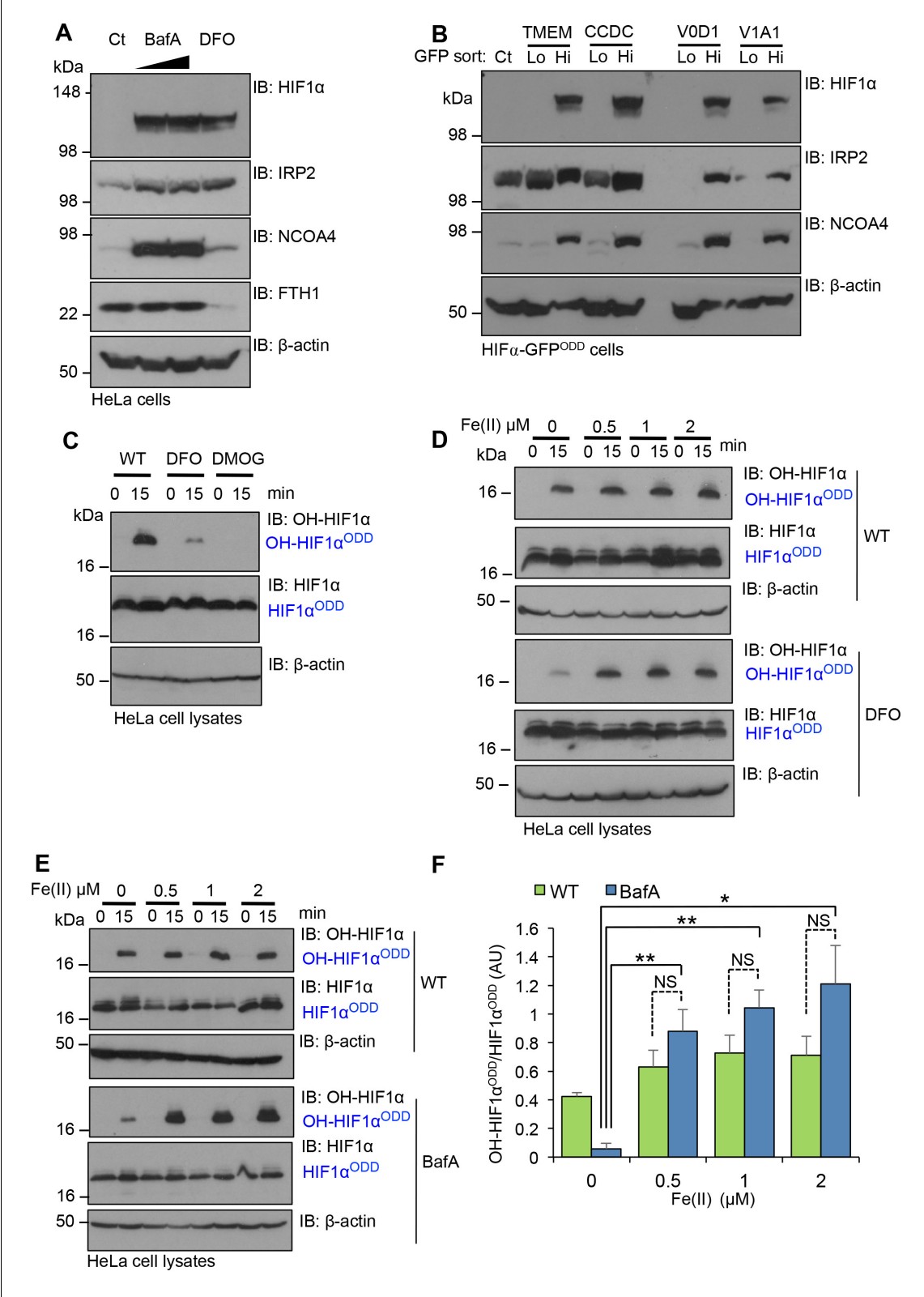

**Figure 8.** Disrupting V-ATPase activity decreases intracellular iron levels. (**A, B**) V-ATPase inhibition leads to intracellular iron depletion. (**A**) HeLa cells were treated with BafA (10 nM or 100 nM) or 100 μM DFO for 24 hr. HIF1α, IRP2, NCOA4 and ferritin (ferritin heavy chain 1, FTH1) levels were measured by immunoblot. (**B**) HIF1α-GFP$^{ODD}$ reporter cells transduced with Cas9 and sgRNA targeting V-ATPase components (TMEM199, CCDC115, ATP6V0D1 and ATP6V1A1) were sorted into GFP$^{LOW}$ (Lo) and GFPHIGH (Hi) populations as described. The lysates were immunoblotted for HIF1α, IRP2, or

*Figure 8 continued on next page*

*Figure 8 continued*

NCOA4. β-actin served as a loading control. (C) Iron chelation prevents HIF1α hydroxylation. In vitro prolyl hydroxylation of the HIF1α^ODD protein following incubation with lysates from WT or DFO treated lysates (100 µM for 24 hr) as previously described. DMOG served as a control for PHD inhibition. (D–F) In vitro hydroxylation of PHD activity in DFO or BafA treated lysates supplemented with ferrous iron. Lysates from control, DFO (D) or BafA (E) treated HeLa cells were extracted as previously described, incubated with the HIF1α^ODD protein, and supplemented with increasing concentrations of iron chloride (FeCl$_2$, Fe(II)). Prolyl-hydroxylated HIF1α^ODD levels were visualised by immunoblot and quantified by densitometry for the BafA treated lysate (F) (n = 3). Values are mean±SEM. *p<0.05, **p<0.01 Fe(II) compared to no treatment in BafA treated cells. NS=not significant.

mechanisms for reduced PHD activity when the V-ATPase is inhibited, and highlight the relative importance of iron uptake versus release from intracellular stores. Interestingly, Fe(III) supplementation to the media was sufficient to restore HIF1α turnover when the V-ATPase was inhibited or following depletion of the transferrin receptor. Thus, transferrin-independent uptake of iron can occur when there is sufficient extracellular iron availability, presumably by transporters such as Zip14, which mediate non-transferrin bound iron uptake (*Liuzzi et al., 2006*). It will be of interest to explore the biological implications of transferrin-independent mechanisms and the role of ferritinophagy in future studies. Moreover, it is plausible that HIF activation and V-ATPase activity serve as a feedback mechanism to control intracellular iron, particularly as HIFs promote genes involved in iron metabolism (*Pantopoulos et al., 2012*; *Peyssonnaux et al., 2008*; *Simpson and McKie, 2015*).

The identification of TMEM199 and CCDC115 as V-ATPase assembly proteins has interesting implications for our understanding of how the human V-ATPase complex forms. In yeast, Vma12p and Vma22p dimerise and are thought to promote the assembly of the membrane embedded complex (V0), which can subsequently associate with the peripheral complex (V1) (*Graham et al., 1998*; *Hill and Stevens, 1995*; *Hirata et al., 1993*; *Jansen et al., 2016c*). Similarly to the yeast Vma12 studies, we find that TMEM199 is predominantly localised to the endoplasmic reticulum, implying that it is likely to be involved in V-ATPase assembly, rather than forming part of the mature complex. Human germline mutations in TMEM199 and CCDC115 cause disorders with a liver storage disease phenotype (*Jansen et al., 2016b*, *2016c*) and it remains to be determined how impaired glycosylation relates to V-ATPase function, or if these defects relate to other roles of TMEM199 and CCDC115. However, it is of interest that all described TMEM199 and CCDC115 human mutations have impaired glycosylation of transferrin, suggesting a role for the V-ATPase associated factors in iron metabolism (*Jansen et al., 2016b*, *2016c*). ATP6AP1 (also known as Ac45) was also identified in our forward genetic screen and has recently been identified as the human orthologue of the yeast Voa1 V-ATPase assembly factor (*Jansen et al., 2016a*). Human mutations in ATP6AP1 have a distinct phenotype to that of TMEM199 and CCDC115 mutations, resulting in immunodeficiency as well as abnormal glycosylation (*Jansen et al., 2016a*). Furthermore, human mutations in another V-ATPase assembly factor, VMA21, cause an autophagic myopathy (*Ramachandran et al., 2013*). The explanation for the diverse nature of human diseases caused by mutations in V-ATPase assembly factors is unclear. Moreover, while yeast studies support the role of Vma12p, Vma21p and Vma22p in V-ATPase assembly, this has not been studied in mammalian cells, where they may serve additional regulatory functions on V-ATPase activity. Addressing the pathological role of these human V-ATPase mutations on iron metabolism and HIFs in animal models will be important in future studies. Nevertheless, our findings show that V-ATPase activity can alter HIF signalling through regulating intracellular iron.

## Materials and methods

### Cell culture, antibodies and reagents

HeLa, HEK293T and RCC10 cells were cultured and maintained at 37°C with 5% (v/v) CO$_2$ in Dulbecco's modified Eagle Medium supplemented with 10% (v/v) Fetal Calf Serum (HyClone) and 100IU/ml Penicillin G and 100 µg/ml Streptomycin. Primary dermal fibroblasts (Lonza) were cultured under the same conditions but with 20% (v/v) FCS. ATG16 null HeLa cells were a kind gift from David Rubinsztein (University of Cambridge). HeLa and HEK293 cells were originally a gift from Paul Lehner (University of Cambridge). RCC10 cells were a gift from Patrick Maxwell (University of Cambridge). HeLa, HEK293 and RCC10 cells were authenticated by STR profiling (Eurofins Genomics). Human

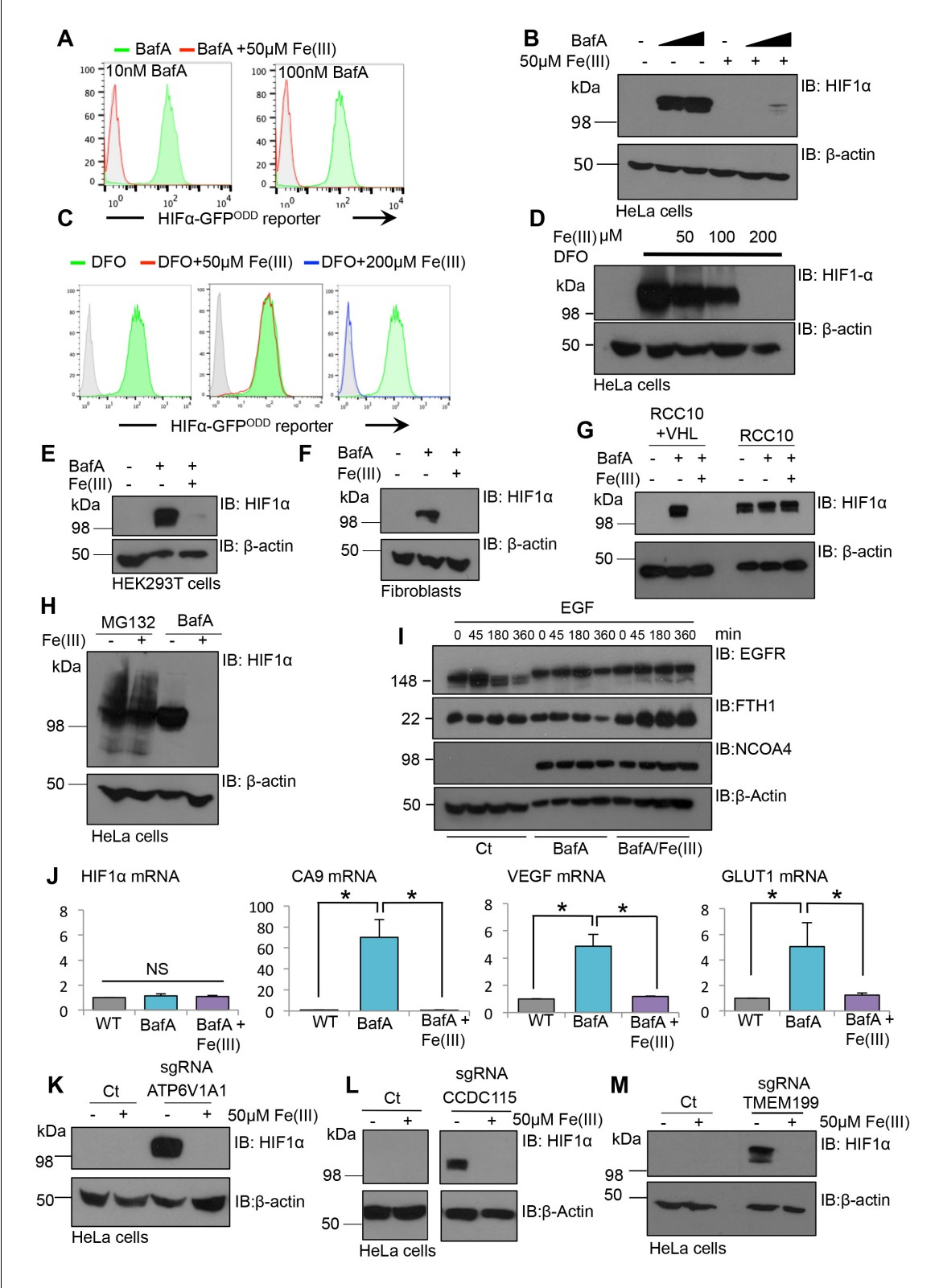

**Figure 9.** Iron supplementation restores HIF1 activity to basal levels following V-ATPase inhibition in cell lines and primary cells. (A–D) Iron reconstitution in BafA or DFO treated HeLa cells. (A, C) HIF1α-GFP^ODD reporter cells were treated with BafA (10 nM or 100 nM), or 100 μM DFO for 24 hr with 50 μM iron citrate (Fe(III)) (red), 200 μM Fe(III) (blue) or no iron (green), and GFP levels analysed by flow cytometry. (B, D) Wildtype HeLa cells were treated with or without BafA (10 nM or 100 nM) or DFO (100 μM) and Fe(III) (50–200 μM) as described, and endogenous HIF1α levels were

*Figure 9 continued on next page*

*Figure 9 continued*

measured by immunoblot. (**E–G**) HEK293T cells (**E**), human dermal fibroblasts (**F**) and RCC10 VHL null and VHL reconstituted cells (**G**) were treated with BafA (10 nM) with or without the addition of 50 μM iron citrate (Fe(III)). HIF1α levels were visualised by immunoblot. β-actin served as a loading control. (**H**) HeLa cells were treated with 20 μM MG132 for 2 hr or 10 nM BafA for 24 hr with or without the addition of 50 μM iron citrate. (**I**) EGFR degradation assay for BafA treated cells following iron treatment. HeLa cells were cultured with 10 nM BafA for 24 hr, with or without 50 μM iron citrate (Fe(III)), and stimulated with EGF as previously described. EGFR, NCOA4, and ferritin (FTH1) levels were visualised by immunoblot. β-actin was used as a loading control. (**J**) RT-qPCR analysis of HIF1α and its target genes in response to BafA and iron citrate treatment (n ≥ 2). (**K–M**) Populations of mixed CRISPR KO cells for ATP6V1A1 (**K**), TMEM199 (**L**) and CCDC115 (**M**) were treated with 50 μM iron citrate for 24 hr and HIF1α levels measured by immunoblot. Values are mean±SEM. *p<0.05, **p<0.01. NS = not significant.

primary dermal fibroblasts were authenticated by Lonza. All cells were confirmed to be mycoplasma negative (Lonza MycoAlert).

Primary antibodies were prepared for immunoblotting as follows: TMEM199 (Atlas, HPA027051, 1:2000), LAMP2A (Abcam, ab18528, 1:1000), HSC70 (Abcam, ab19136, 1:1000), EGFR (Santa Cruz, Sc-03, 1:500), HIF1α (BD Transduction Laboratories, 610959, 1:1000), β-actin (Sigma, A228, 1:30000), Calnexin (Abcam, ab75801, 1:1000), Tubulin (eBioscience 14–4502, 1:1000), ATG16L (MBL, PMO40, 1:1000, Gift from Rubinzstein lab, CIMR), M2-FLAG (Sigma, F3165, 1:5000), CCDC115 (Atlas, HPA034598, 1:1000), PHD2 (Novus bioscience, NB100-137, 1:5000), Hydroxy-HIF-1α (Cell signaling technology, 3434, 1:2000), β2M (Dako, A00072, gift from Paul Lehner, CIMR, 1:10000), NCOA4 (ARA70, Bethyl Laboratories, A302-272A-T, 1:2000), IRP2 (IREB2, Cell Signalling Technology, 37135, 1:1000), Ferritin heavy chain (FTH1, Cell Signalling Technology, 3998, 1:1000), Carbonic anhydrase 9 (CA9, clone M75, gift from Egbert Oosterwijk, Radboud, 1:5000). Immunofluorescence antibodies: LAMP1 (Santa Cruz, SC-20011, 1:100), TGN46 (AbD Serotec, AHP500GT, 1:200), KDEL (gift from Geoff Butcher, Babraham Institute, 1:400), EEA1 (gift from Evan Reid, 1:250), M6PR (gift from Evan Reid, 1:500), HIF1α (1:100), TMEM199 (1:50), HA-11 (Ms Covance, MMS-101P, 1:100), rabbit polyclonal to HA-11 (gift from Paul Lehner, 1:100). Antibodies for cell-surface staining: MHC Class I (W6/32, gift from Paul Lehner, CIMR, 1:100), Transferrin receptor (TFRC, CD71, BD Pharmigen, 555534, 1:500), CA9 (1:500).

The following reagents were used: MG132 (Sigma-Aldrich, 20 μM), Bafilomycin A1 (Alfa Aesar, J61835, 10 nM-100nM), Ammonium Chloride (Sigma-Aldrich, A9434, 10 mM), DMOG (Sigma-Aldrich, D3695, 0.5 mM), Chloroquine (Sigma-Aldrich, C6628, 50 μM), DFO (Sigma-Aldrich, D9533, 100 μM). Puromycin, Hygromycin, and Blasticidin were purchased from Cambridge Bioscience (all used at 10 μg/ml). ProLong Gold Antifade Reagent with DAPI (8961, Cell signaling technology, 8961).

## Plasmids

The following plasmids were used. LentiCRISPR v2 (sgRNA/Cas9, F. Zhang Addgene #52961), pHRSIN-pSFFV-FLAG-MPP8-pPGK-Blasto (Gift from Paul Lehner), TMEM199 Image Clone (Source Bioscience). CCDC115 Image Clone (Source Bioscience), pHRSIN-pSFFV-HA-Ube2J2 -pPGK-Puro (gift from Paul Lehner). pMD.G (Lentiviral VSVG), pMD.GagPol (lentiviral Gag/Pol). Lentiviral plasmids used the pHRSIN backbone (*Demaison et al., 2002*). The HIF1α-GFP[ODD] reporter was generated as described (*Burr et al., 2016*). The HIF1α-mCherry[ODD] reporter was generated by excising the GFP and subcloning mCherry CL1 using the BamHI and NotI restriction sites. Tfnr-phl was a gift from Christien Merrifield (*Merrifield et al., 2005*).

## Forward genetic screen and bioinformatic analyses

The KBM7 forward genetic screen was carried out as previously described (*Burr et al., 2016*). Bioinformatic pathway analyses were performed by taking genes enriched for trapping insertions with an adjusted Fisher Exact test p-value<0.1 and running an 'express analysis' using Metascape 1.0 (*Tripathi et al., 2015*). All statistically enriched terms were identified (accumulative hypergeometric p-values and enrichment factors were calculated and used for filtering), and were hierarchically clustered into a tree based on Kappa-statistical similarities among their gene memberships. Then 0.3 kappa score was applied as the threshold to cast the tree into term clusters. We selected a term

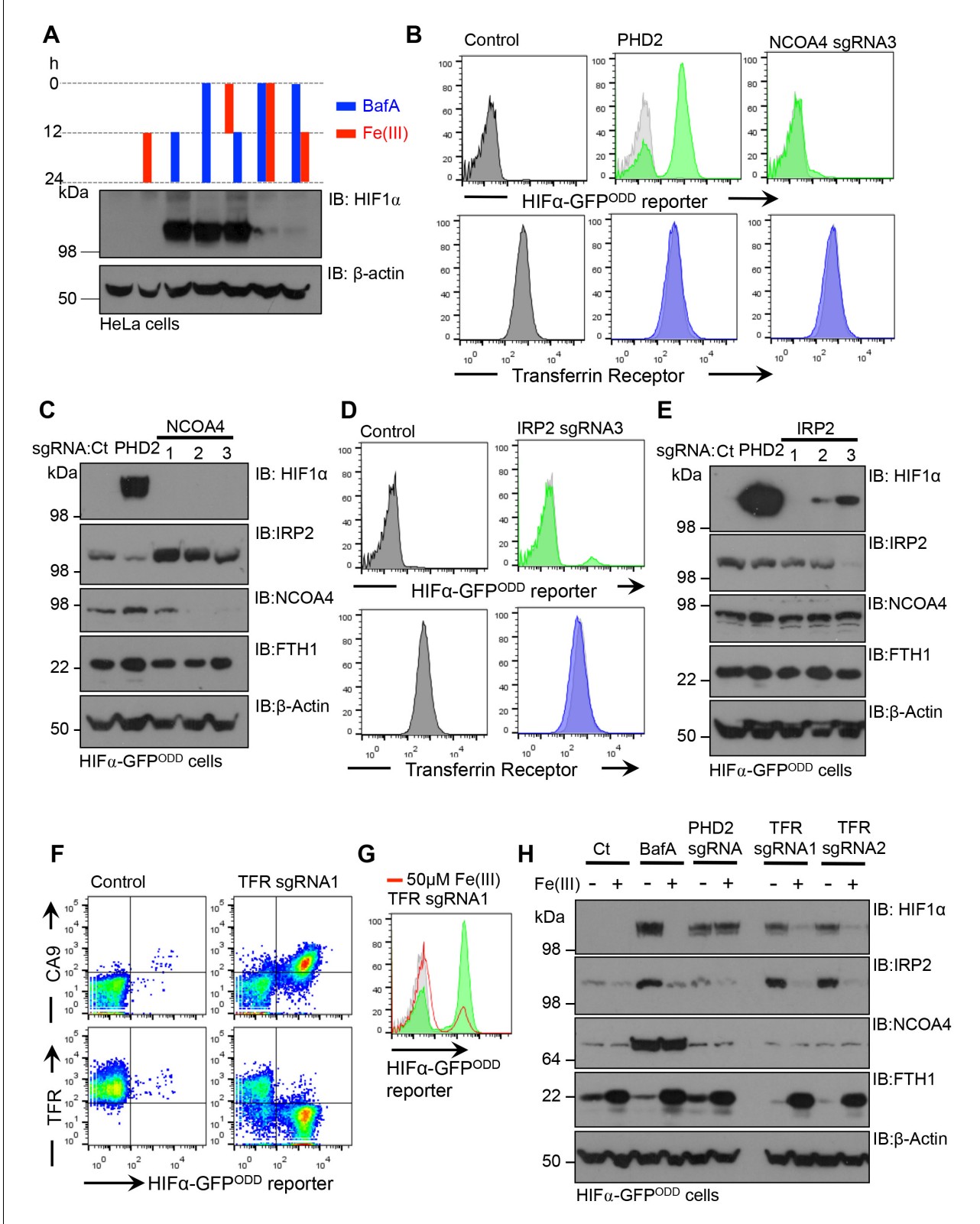

**Figure 10.** Disrupting transferrin uptake leads to iron-dependent HIF1 activation. (**A**) The effect of iron treatment on HIF1α stabilisation in the presence or absence of BafA. HeLa cells were treated with 50 μM iron citrate (red) or 10 nM BafA (blue) for the indicated times and HIF1α levels measured by immunoblot. β-actin served as a loading control. (**B–E**) The effect of NCOA4 or IRP2 depletion on HIF1α levels and intracellular iron levels. HIF1α-GFP[ODD] reporter HeLa cells were transduced with Cas9 and sgRNAs to NCOA4 (**B, C**) or IRP2 (**D, E**) and HIF1α levels measured by GFP accumulation

*Figure 10 continued on next page*

*Figure 10 continued*

(B, D) top) and immunoblot after 8 (IRP2) or 12 days (NCOA4) (C, E). Cell surface transferrin levels were measured by flow cytometry (B, D) bottom). IRP2 and ferritin (FTH1) levels were visualised by immunoblot (C, E). An sgRNA to PHD2 was used as a control. (F) HIF1 activation following depletion of the transferrin receptor. HIF1α-GFP$^{ODD}$ reporter HeLa cells were transduced with Cas9 and sgRNAs to the transferrin receptor. HIF1 activation was measured by GFP accumulation and cell surface expression of CA9 (top). Depletion of the transferrin receptor was measured by flow cytometry (bottom) (G, H) Iron reconstitution restores HIF1α turnover in transferrin receptor deficient cells. HIF1α-GFP$^{ODD}$ reporter HeLa cells were transduced with Cas9 and sgRNAs to the transferrin receptor as described, and 50 μM ferrous citrate added to media for 24 hr. HIF1α levels were measured by flow cytometry for GFP fluorescence (G) or immunoblot (H). IRP2, NCOA4 and FTH1 were visualised to measure intracellular iron. β-actin served as a loading control. Additional experimental examples with alternative sgRNAs to NCOA4, IRP2 or the transferrin receptor are shown in *Figure 10—figure supplement 1*. TFR=transferrin receptor.

The following figure supplement is available for figure 10:

**Figure supplement 1.** Disrupting transferrin uptake leads to iron-dependent HIF1 activation.

---

with the best p-value as the representative term for each cluster and show the p-values of all clusters in a bar graph.

Structures of CCDC115 and Vma22p were predicted using the Protein Homology/Analogy Recognition Engine v2.0 (Phyre$^2$) (*Kelley et al., 2015*). These were then imported into the PyMOL Molecular Graphics System (Schrödinger LLC). Images were rendered using the 'cartoon' function and aligned using the 'align' function.

## Lentiviral production and transduction

Lentivirus was prepared by transfection using Mirus Trans-IT−293 Transfection Reagent in Hek293ET cells. Cells were triple transfected with pMD.G (VSVG envelope), the suitable lentiviral transgene vector and pCMVR8.91 (gag/pol) at a ratio 2:3:4. Transfection was performed in six well plates at 70–80% confluency. The resultant viral supernatant was harvested at 48 hr, filtered through a 0.45 μm filter and stored at −80°C. To achieve stable integration by transduction, cells were seeded to 24 well plates in 500 μl DMEM. Viral supernatant (500 μl) was added to each well and plates spun at 1800 rpm for 1 hr at 37°C. Plates were incubated post spin and selected for antibiotic resistance after 48 hr.

## Flow cytometry

Cells were harvested, washed in PBS before fixing in 1% paraformaldehyde, PBS and analysed on a FACSCalibur (BD). For FACS, cells were harvested, washed in 10 mM Hepes PBS and re-suspended in sorting medium (10 mM Hepes, 2% FCS, PBS). Cell suspensions were filtered prior to sorting in a High speed Influx Cell Sorter (BD Biosciences). For cell surface staining, cells were washed in PBS and incubated with the appropriate primary antibody for 30 min at 4°C. After a further PBS wash the cells were incubated with secondary antibody for 30 min at 4°C. A final PBS wash was performed and cells were fixed in PBS with 1% paraformaldehyde (PFA) and analysed as described.

## CRISPR-Cas9 targeted deletions

Gene-specific CRISPR sgRNA oligonucleotide sequences were selected using the GeCKO v2 library. Sense and antisense sgRNAs oligonucleotides were designed with 5' CACC and 3' CAAA overhangs respectively. The subsequent sgRNAs were cloned into the LentiCRISPRv2 backbone (*Sanjana et al., 2014*) or pKLV-U6gRNA(BbsI)-PGKpuro2ABFP. The following sgRNAs were used:

| | |
|---|---|
| TMEM199 | TATGGCGTCCTCTTTGCTTG |
| ATP6V0D1 | TCGATGACTGACACCGTCAG |
| ATP6AP1 | GCTGACTGCATACCAGTCGA |
| ATP6V1A1 | GTAACTTACATTGCTCCACC |
| ATP6V0A2 | GCGACACTCACGTCTCGGAAC |
| ATP6V1G1 | GTGAAAACAGGAAAGAACCG |

| | |
|---|---|
| B2M | GGCCGAGATGTCTCGCTCCG |
| PHD2 | ATGCCGTGCTTGTTCATGCA |
| CCDC115 | GGGGGCTCACCTGCTTCGCG |
| HSC70 sgRNA1 | ACCATAGAAGACACCTCCTC |
| HSC70 sgRNA2 | CTAGACTGTTACCAATGCTG |
| HSC70 sgRNA 3 | GACAGATGCCAAACGTCTGAT |
| LAMP2A sgRNA1 | ACCAGAACGAGCCCTGAGCC |
| LAMP2A sgRNA2 | TCCGGGCTCAGGGCTCGTTC |
| LAMP2A sgRNA3 | CAAGAACATCCCAGTAGTGT |
| NCOA4 sgRNA1 | GGTATGGCTGTATGAACAGG |
| NCOA4 sgRNA2 | CAATCTCCACACCTTTGGGC |
| NCOA4 sgRNA3 | TAGCTGTCCCTTTCAGCGAA |
| IRP2 sgRNA1 | AATGCACCAAATCCTGGAGG |
| IRP2 sgRNA2 | TGAGCCATTCCAGTTCCAGG |
| IRP2 sgRNA3 | GCATAAGCTACCACTAAGGG |
| Transferrin Receptor sgRNA1 | AAAGTCTCCAGCACTCCAAC |
| Transferrin Receptor sgRNA2 | GCTCTGGAGATTGTCTGGAC |

## siRNA-mediated depletion

100 nM HSC70 siRNA (Dharmacon Smartpool, L-017609-00-0005) or MISSION siRNA Universal Negative Control (control) was transfected into $3 \times 10^5$ HeLa cells using Oligofectamine Transfection Reagent (Invitrogen) according to the manufacturer's protocol. Cells were harvested after 96 hr for further analysis by immunoblot.

## Immunofluorescence

Cells were cultured on glass coverslips, washed in PBS and fixed in 4% (w/v) PFA, PBS at room temperature for 20 min. Cells were then blocked and permeabilised in 3% BSA, 0.3% TritonX-100, PBS. For observing endosomal compartments, cells were permeabilised in 3% BSA, 0.05% Saponin, PBS. Coverslips were incubated with primary antibody for 1 hr, washed in PBS and fluorophore-conjugated secondary antibodies applied for 30 min. Coverslips were mounted to microscope slides using ProLong Gold antifade with DAPI. Imaging was performed on a Zeiss LSM880 inverted confocal microscope.

Cytoplasmic colocalisation analyses of stains were performed using CellProfiler image analysis software (Carpenter et al., 2006). The analysis was performed blinded. Nuclear and cell boundaries were identified manually, with the area between the nuclear and cell peripheries classified as the cytoplasm. The 'Measure Correlation' module was used to calculate the Pearson's Correlation Coefficient between the pixel intensities for the described stains.

For the Tfnr-phl experiments, wildtype or HIF1α-mCherry$^{ODD}$ reporter HeLa cells were transfected with TfR-pHluorin using TransIT-HeLaMONSTER Transfection kit (Mirus Bio LLC) according to manufacturer's protocol. To perform fixed cell immunofluorescence, cells were cultured on glass coverslips, washed in PBS and fixed in 4% (w/v) PFA, PBS as previously described. Cells were permeabilised in PBS with 3% BSA, 0.05% Saponin, and coverslips mounted to microscope slides using ProLong Gold antifade with DAPI. Imaging was performed on a Zeiss LSM880 confocal microscope. Live cell imaging was performed on a Zeiss LSM780 inverted confocal microscope equipped with a 63x objective. To examine Tfnr-phl localization cells were treated with or without 10 nM BafA for 24 hr prior to imaging. pH clamping was performed during live cell imaging by incubating cells with 25 mM sodium acetate buffer (pH 5), 25 mM MES buffer (pH 6) and 25 mM Hepes buffer (pH 7) containing 5 mM NaCl, 1 mM CaCl$_2$, 115 mM KCl, 1.2 mM MgSO$_4$, 10 mM glucose, 10 μM nigericin and 10 μM monensin for 5 min at 37℃ as previously described (Bright et al., 2016). Fluorescence intensity measurements of Tfnr-phl were performed using CellProfiler image analysis software (Carpenter et al., 2006). The analysis was performed blinded. Cell surface and cytoplasmic

boundaries were identified manually and the ratio between cytoplasmic fluorescent intensity and whole cell fluorescence was calculated.

## Immunoblotting

Cells were lysed in SDS lysis buffer (1% SDS, 50 mM Tris pH7.4, 150 mM NaCl, 10% glycerol and 5µl/ml benzonase nuclease) for 10 min on ice, heated at 90°C for 5 min and centrifuged at 14,000 rpm for 10 min. Proteins were separated by SDS-PAGE, transferred to methanol activated Immobilon-P 0.45 µm PVDF membrane, probed with appropriate primary and secondary antibodies, and developed using SuperSignal West Pico or Dura Chemiluminescent Substrates (Thermo Scientific).

## Membrane fractionation

HeLa cells ($3 \times 10^6$) were harvested, washed and resuspended in 1 ml break buffer (20 mM Hepes pH7.4, 1 mM EDTA, 0.5 mM $MgCl_2$, 0.13 M sucrose, 50 mM NaCl, 1 mM PMSF supplemented with Roche complete EDTA free protease inhibitors), and passed through an equilibrated ball bearing homogenizer at 1 µm diameter for 20 passes. The homogenized lysate was collected and centrifuged at 3000 rpm for 10 min to pellet nuclei. The post nuclear supernatant was ultracentrifuged at 50,000 rpm for 1 hr to pellet intracellular membranes before resuspension in SDS loading buffer for separation by SDS-PAGE and immunoblot.

## EGFR degradation assay

This assay was performed as previously described (*Almeida et al., 2006*), with some modifications. HeLa cells were cultured in the presence or absence of BafA on six well plates prior to serum starvation for 90 min in DMEM supplemented with 2% FCS and cyclohexamide (100 µg/ml). Following starvation, cells remained in 2% FCS and were stimulated with 100 ng/ml EGF (Cabiochem). The reaction proceeded for 0, 10, 45, 90 and 180 min until quenched on ice. The cells were washed in ice cold PBS and lysed on ice in SDS lysis buffer. EGFR expression levels were probed by immunoblotting.

## Immunoprecipitation and mass spectrometry coupled immunoprecipitation

HeLa cells ($5 \times 10^6$) were lysed in 1% NP-40, TBS supplemented with 1x Roche cOmplete EDTA-free protease inhibitor cocktail for 30 min at 4°C before centrifugation at 14,000 rpm for 10 min. The supernatants were then diluted to 0.1% detergent and pre-cleared with SP sepharose Fast-Flow (GE Healthcare) for 1 hr. The pre-cleared supernatant was incubated with 40 µl anti-FLAG M2 magnetic beads (M8823, Sigma Aldrich) for 3 hr at 4°C. The resins were then washed and the bound proteins eluted with 100 µg/ml FLAG peptide (30 min). The eluted proteins were then separated by SDS-PAGE and immunoblotted as described.

For mass spectrometry analysis of TMEM199 associated proteins, wildtype or TMEM199 KO HeLa cells ($1 \times 10^8$) were immunoprecipitated with the TMEM199 antibody conjugated to Dynabeads Protein G for 3 hr at 4°C. The resins were then washed and sample eluted with SDS lysis buffer at 70°C for 10 min. Samples were resolved a short distance into an SDS-polyacrylamide gel, the lanes excised and subjected to in-gel tryptic digestion. The resulting peptides were analysed using a Q Exactive (Thermo Scientific) coupled to an RSLC3000nano UPLC (Thermo Scientific). Files were searched against a Uniprot human database (downloaded 09/06/14, 20,264 entries) using Mascot with peptide and protein validation performed in Scaffold.

## Quantitative real-time PCR

Total cellular RNA was isolated and purified using the Qiagen RNeasy Plus Mini Kit (Qiagen, UK) according to manufacturer's protocol followed by reverse transcription using SuperRT (HT Biotechnology Ltd). PCR reactions (15 µl) were prepared using SYBR Green PCR Master Mix (Applied Biosystems) with 125 ng starting cDNA template. The reaction proceeded in an ABI 7900 HT Real-Time PCR system (Applied Biosystems) and the resultant Ct values were normalized to housekeeping genes (GAPDH and RPS2). The following primers were used:

GAPDH F:ATGGGGAAGGTGAAGGTCG R: CTCCACGACGTACTCAGCG

HIF1α F: CCAGTTACGTTCCTTCGATCAGT R: TTTGAGGACTTGCGCTTTCA

GLUT1 F: TGGCATGGCGGGTTGT R: CCAGGGTAGCTGCTCCAGC
VEGF F: TGCCAAGTGGTCCCAG R: GTGAGGTTTGATCCGC

## In vitro hydroxylation assay

Prolyl hydroxylation of the HIF1$\alpha^{ODD}$ protein was performed as described in *Burr et al. (2016)*. Briefly, the hydroxylation assay was performed by incubating 10 µM HIF1$\alpha^{ODD}$ with 50 µl HeLa cell extract for 15 min at 37°C. The reaction was stopped by addition of SDS loading buffer, and the proteins separated by SDS-PAGE. Hydroxylation was measured using the HIF prolyl hydroxylation specific antibody. Measurements of HIF1$\alpha$ hydroxylation following the addition of Fe(II) were performed similarly, except the lysate was pre-incubated with iron chloride for 10 min at 4°C before the addition of the HIF1$\alpha^{ODD}$ protein.

## Statistical analyses

Data were expressed as mean ± s.e.m. and P values were calculated using two-tailed Student's t-test for pairwise comparisons, unless otherwise stated. The cytofluorometric colocalisation studies were analysed as described and performed blinded. No statistical method or power analysis was used to predetermine sample size.

## Acknowledgements

ALM, SPB, GLG and JAN designed the studies and performed the experiments. ALM, SPB and JAN wrote the manuscript. We thank Ian Lobb for assistance with quantification of the fluorescence microscopy and generating the HIF1$\alpha$-mCherry$^{ODD}$ construct, and Peter Sterk with the bioinformatics analysis. We also thank Paul Luzio, Nick Bright, Luther Davis, Symeon Siniossoglou, and the Nathan laboratory for their helpful discussions. This work was supported by a Wellcome Trust Senior Clinical Research Fellowship to JAN (102770/Z/13/Z), and an MRC Award to ALM (MR/K50127X/1). The Cambridge Institute for Medical Research is in receipt of a Wellcome Trust Strategic Award (100140).

## Additional information

### Funding

| Funder | Grant reference number | Author |
| --- | --- | --- |
| Wellcome | 102770/Z/13/Z | Stephen P Burr<br>Guinevere L Grice<br>James A Nathan |
| Medical Research Council | MR/K50127X/1 | Anna L Miles<br>James A Nathan |

The funders had no role in study design, data collection and interpretation, or the decision to submit the work for publication.

### Author contributions

ALM, SPB, Conceptualization, Data curation, Formal analysis, Validation, Investigation, Visualization, Methodology, Writing—original draft, Project administration, Writing—review and editing; GLG, Validation, Investigation, Visualization, Methodology; JAN, Conceptualization, Resources, Data curation, Formal analysis, Supervision, Funding acquisition, Investigation, Writing—original draft, Project administration, Writing—review and editing

### Author ORCIDs

James A Nathan, http://orcid.org/0000-0002-0248-1632

## Additional files

**Supplementary files**
• Supplemental file 1. List of genes enriched in the KMB7 genetic screen ranked according to biological process and function.

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
