## [Decision Letter]

[Editors’ note: this article was originally rejected after discussions between the reviewers, but the authors were invited to resubmit after an appeal against the decision.]

Thank you for submitting your work entitled "The vacuolar-ATPase complex and assembly factors control HIF1α hydroxylation by regulating cellular iron levels" for consideration by *eLife*. Your article has been favorably evaluated by a Senior Editor and three reviewers, one of whom is a member of our Board of Reviewing Editors. The reviewers have opted to remain anonymous.

Our decision has been reached after consultation between the reviewers. Based on these discussions and the individual reviews below, we regret to inform you that your work will not be considered further for publication in *eLife*.

As you will see from the individual comments of the Referees appended below, the findings of this work were found interesting but too preliminary. Importantly, links with the systems responsible for iron metabolism and the characterization of putative V-ATPase assembly factors and/or physio-pathological consequences should be expanded. Addressing these issues will likely exceed the time span for major revisions in *eLife*.

*Reviewer #1:*

The manuscript characterizes a link between the HIF1α pathway and the endo-lysosomal machinery. Based on a genetic screen the authors identify several subunits of V-ATPase to control degradation of HIF1α under normoxic conditions. The authors correlate the activity of prolyl hydroxylases, which flag HIF1α for degradation by the ubiquitin-proteasome system, with the import of iron provided by the endocytic pathway. The manuscript reports that supplementation of iron rescues stabilization of HIF1α induced both by chemical (bafilomycin A) and genetic inhibition of V-ATPase. In addition to mere stabilization of HIF1α, the authors demonstrate the activation of HIF1 target genes proving that the inhibition of V-ATPase can trigger a hypoxic program.

The manuscript presents a suggestive set of data in favor of authors' conclusion, yet it lacks full verification of the claims. The effect of iron depletion on HIF1α by a means that is independent on inhibition or depletion of V-ATPase should be determined. In addition, the authors should provide more evidence that HIF1α stabilization due to V-ATPase inhibition is mediated by PHD activity, and not by the increase in mRNA levels, or other mechanisms leading to the accumulation of non-hydroxylated pool of HIF1α. The existing data linking iron and PHD activity should be presented.

The authors identified the novel assembly factors of V-ATPase TMEM199 and CCDC115, which are orthologues of yeast Vma12p and Vma22p and similarly to V-ATPase subunits influence the stability of HIF1α protein. The authors confirm interaction between both proteins via co-immunoprecipitation, verify their cellular localization and show that V-ATPase functionally depends on these proteins. Thus, they are good candidates for assembly factors. To fully support authors' conclusion, the integrity of V-ATPase could be addressed via Western blot and immunoprecipitation. To strengthen the link between V-ATPase biogenesis and function, the uptake of iron by the cells depleted of TMEM199 and CCDC115 should be determined.

*Reviewer #2:*

In this manuscript the authors have screened for cellular proteins that contribute to the degradation of HIF1 α under conditions of oxygen sufficiency. They identify components of the lysosomal V-ATPase and two uncharacterized proteins that may function as V-ATPase assembly factors. Examination of the activity of the prolyl hydroxylase that mediates the HIF hydroxylation that leads to degradation reveals that when the V-ATPase is inhibited, PHD activity is reduced. They trace this loss of PHD activity to cytosolic depletion of the iron cofactor.

The authors have taken a relatively important biological question (how is the HIF transcription factor regulated) and used a broad, relatively low-bias, forward genetic screen (gene trapping in a near-haploid cell line) to identify genes involved in HIF degradation in aerobic conditions. What they found (components of the V-ATPase) is fairly well-supported by the evidence and the mechanism they propose (impaired iron uptake) makes sense biologically.

I have two major concerns with these studies in their current form.

First, the evidence that TMEM199 and CCDC115 are V-ATPase assembly factors is not as strong as it could be. Several approaches could be useful in strengthening this aspect of the manuscript. The authors present a single assay of lysosomal function, the degradation of EGFR. While these data are consistent with their hypothesis, it would be strengthened if additional evidence were presented. Is the acidification of the lysosome impaired in the absence of these factors? A pH-sensitive fluorescent dye is available and could easily be used here. Will the yeast homologues of these proteins substitute for their human counterparts in a human cell lacking TMEM or CCDC? Are any proteins (other than EGFR) that are turned over in the lysosome and are also affected by depletion of these proteins?

A second concern is that the authors attribute the loss of PHD activity to a loss of iron uptake in the V-ATPase-deficient cells. Cytosolic iron levels are maintained by uptake, yes, but also by the balance between storage and mobilization of iron in ferritin. Ferritin is turned over in the lysosome by an autophagic process that requires the cargo receptor NCOA4 and an appropriately acidified lysosome. While the authors show that increased iron uptake after external iron supplementation will restore cytosolic iron levels and lead to PHD activation, the defect associated with the V-ATPase may not be defective iron uptake, but defective iron recycling from ferritin. The rapid onset of PHD inactivation, even in cells pre-loaded with iron (Figure 6), suggests this may be the case. Drugs that inhibit the lysosomal V-ATPase, e. g., bafilomycin A, will also inhibit the acidification of the endosome, so these drugs don't really distinguish between impaired uptake of iron and impaired turnover of ferritin. The authors should clarify this experimentally. Iron uptake may be measured directly in assays using stable or radioactive isotopes of iron. Although direct measurement of cytosolic iron levels is technically fraught, as the authors explain, indirect measurements are possible, however. Iron Regulatory Protein 2 is rapidly degraded via the proteasome in iron-replete cells and is a good indicator of cytosolic iron availability. If the author's iron hypothesis is correct, IRP2 levels should increase with inhibition of the V-ATPase. Proteins involved in autophagic delivery of ferritin to the lysosome could be depleted and PHD activity measured. This approach would add additional clarity to the mechanism of iron depletion. As this is the most novel aspect of these studies, it represents an important opportunity for novel and significant observations.

The concerns I outlined above are important, because the role of the two new proteins as assembly factors is far from certain and they should present stronger evidence using more than one experimental approach to support their claim. My second concern is equally important. Investigators in the iron field have not focused on the regulation of ferritin turnover until very recently because the genetic components that carry out the turnover have only recently been identified. The role of ferritin turnover in the lysosome is the potentially novel finding in this study. We already know that PHD is an iron-dependent enzyme and that chelators that block iron uptake will lead to inactivation of PHD. We already know that drugs that inhibit the V-ATPase will inhibit the uptake of Tf-bound iron, but this effect is kind of small in cultured cells unless you treat for many hours. What we don't know is the relative importance of the flow of iron through ferritin and the turnover of ferritin in the lysosome in maintaining cytosolic iron levels and the activity of cytosolic iron enzymes. They should take a look at this before they ascribe their effects to impaired uptake. If the authors choose not to do further studies, then they must acknowledge the ambiguity and inconclusive nature of their results. Furthermore, the impact and significance of these studies will be lessened.

*Reviewer #3:*

Degradative pathways of HIF have been well studied, however precise mechanisms and role in hypoxia mediated cell effects are still not clear. The major pathway by which HIF is regulated is through a proteasomal degradation pathway initiated by PHD-dependent hydroxylation of HIF. However, it has been shown that altering lysosomal and chaperone-mediated autophagy can also lead to HIF activation. The present work through a novel forward genetic screen identified several lysosomal V-ATPases that lead to increased HIF-1α expression. Interestingly, this is not via direct lysosomal degradation pathway that was previously shown. The strength of the manuscript is the novel mechanism downstream of lysosomal function. The authors show that HIF-1α stabilization following V-ATPase disruption or inhibition is via changes in cellular iron levels. It is clear that several stimuli and several different mechanisms can alter HIF-1α expression and activity. This work provides interesting downstream mechanism from lysosomes, but understanding the physiological or pathological context of this pathway in HIF-1α activation is missing. Along the same lines, as the author's state there are several diseases due to V-ATPase mutations, some experimental data would be important to show that the initiation or progression of these disease are related to iron handling and/or HIF-1α activation.

The authors rule out CMA pathways due to no effect following disruption of HSC70 and LAMP2A. However, HSC70 disruption is not efficient. Similarly, LAMP2A disruption is not complete and it is not clear if this is specific for LAMP2A or are all LAMP2 isoforms reduced. The authors must demonstrate that indeed CMA is reduced by showing that degradation of well-characterized targets of CMA are altered.

The authors state, "HIF activation and V-ATPase activity serves as a feedback mechanism to control iron uptake, particularly as HIFs promote iron uptake under conditions of iron deficiency". HIF-2α is the major isoform that leads to iron uptake, if this pathway is an important feedback mechanism for cellular iron response HIF-2α expression and activity should be assessed.

Similar to what is shown following BafA in Figure 6, increase in HIF-1α target gene following ATP6V1a1, TMEM199, and CCDC115 disruption would be important to demonstrate the increase in HIF-1α is functional.

---

## [Author Response]

[Editors’ note: the author responses to the first round of peer review follow.]

*As you will see from the individual comments of the Referees appended below, the findings of this work were found interesting but too preliminary. Importantly, links with the systems responsible for iron metabolism and the characterization of putative V-ATPase assembly factors and/or physio-pathological consequences should be expanded. Addressing these issues will likely exceed the time span for major revisions in eLife.*

*Reviewer #1:*

*[…] The manuscript presents a suggestive set of data in favor of authors' conclusion, yet it lacks full verification of the claims. The effect of iron depletion on HIF1α by a means that is independent on inhibition or depletion of V-ATPase should be determined. In addition, the authors should provide more evidence that HIF1α stabilization due to V-ATPase inhibition is mediated by PHD activity, and not by the increase in mRNA levels, or other mechanisms leading to the accumulation of non-hydroxylated pool of HIF1α. The existing data linking iron and PHD activity should be presented.*

The suggestion to include further data examining the effect of iron depletion on HIF1α was helpful, and we now present several new figures that verify iron depletion increases HIF1α levels. The iron chelator desferrioxamine (DFO) has been extensively used to demonstrate the requirement for ferrous iron in PHD function (e.g. PMID 11292861 and 8260699) (subsection “Disrupting the V-ATPase decreases intracellular iron levels and prevents HIF1α prolyl hydroxylation”, second paragraph). To corroborate existing data we show (i) that DFO treatment prevents HIF1α hydroxylation in ourin vitroassay (New Figure 8), and (ii) DFO treatment in cells stabilises our HIF1α-GFP reporter and endogenous HIF1α (New Figure 9). Importantly, HIF1α stabilisation by DFO is reversed by culturing the cells with iron (New Figure 9), similarly to BafA treatment with iron supplementation.

We demonstrated that HIF1α mRNA levels do not alter following V-ATPase inhibition (Figure 9), and have made this point clearer in the revised manuscript (subsection “Iron supplementation restores HIF1α turnover in cells when V-ATPase activity is disrupted”, first paragraph). Our in vitro and cell assays also clearly show that PHD activity is impaired by V-ATPase inhibition. Importantly, prolyl hydroxylation and HIF1α turnover is restored with iron treatment, which strongly argues against a PHD independent mechanism for HIF1α stabilisation.

*The authors identified the novel assembly factors of V-ATPase TMEM199 and CCDC115, which are orthologues of yeast Vma12p and Vma22p and similarly to V-ATPase subunits influence the stability of HIF1α protein. The authors confirm interaction between both proteins via co-immunoprecipitation, verify their cellular localization and show that V-ATPase functionally depends on these proteins. Thus, they are good candidates for assembly factors. To fully support authors' conclusion, the integrity of V-ATPase could be addressed via Western blot and immunoprecipitation. To strengthen the link between V-ATPase biogenesis and function, the uptake of iron by the cells depleted of TMEM199 and CCDC115 should be determined.*

Assessing the integrity of the V-ATPase by immunoprecipitation is challenging given the lack of available good antibodies for V-ATPase subunits (a well recognised limitation in the field). We have therefore focused on the functional integrity of the V-ATPase when TMEM199 or CCDC115 are depleted, by measuring endosomal acidification. Using a pH sensitive transferrin receptor, with a phlourin attached to the extracellular domain (Tfnr-phl) (PMID 15907472), we now show that TMEM199 and CCDC115 depletion prevents acidification of endosomal compartments, similarly to BafA treatment or depletion of core V-ATPase subunits (New Figure 5 and Figure 5—figure supplement 1). This localisation of Tfnr-phl to neutral compartments also strengthens the link between iron metabolism and the V-ATPase, as the release of iron from transferrin requires a low pH (e.g. PMID 21788477 and 16227996). Please also see comments to Reviewer 2 with regard to iron uptake and studies of iron metabolism.

*Reviewer #2:*

*[…] I have two major concerns with these studies in their current form.*

*First, the evidence that TMEM199 and CCDC115 are V-ATPase assembly factors is not as strong as it could be. Several approaches could be useful in strengthening this aspect of the manuscript. The authors present a single assay of lysosomal function, the degradation of EGFR. While these data are consistent with their hypothesis, it would be strengthened if additional evidence were presented. Is the acidification of the lysosome impaired in the absence of these factors? A pH-sensitive fluorescent dye is available and could easily be used here. Will the yeast homologues of these proteins substitute for their human counterparts in a human cell lacking TMEM or CCDC? Are any proteins (other than EGFR) that are turned over in the lysosome and are also affected by depletion of these proteins?*

We have now included several assays to confirm that TMEM199 and CCDC115 are involved in V-ATPase function:

1) We have used a fluorescent pH sensitive transferrin receptor construct (Tfnr-phl) to measure acidification of endosomal compartments when the V-ATPase is inhibited or depleted (see also comments to reviewer 1). We find that TMEM199 and CCDC115 depletion prevent acidification of endosomal compartments, similarly to BafA treatment or depletion of core V-ATPase subunits (New Figure 5 and Figure 5—figure supplement 1). Thus, both TMEM199 and CCDC115 are required for endosomal acidification by the V-ATPase.

2) To further examine the degradative consequences of V-ATPase inhibition or disruption we measured lysosomal degradation of MHC Class I. The KSHV gene product, K3, ubiquitinates MHC Class I at the cell surface, facilitating Class I clathrin-mediated endocytosis and lysosomal degradation. This pathway has been extensively studied and provides a robust experimental model to examine lysosomal degradation (PMID 12006494, 10799607 and 8260699). Using K3 expressing HeLa cells we show that TMEM199 and CCDC115 depletion increases MHC Class I expression at the cell surface and prevents Class I degradation (New Figure 4). We also observe similar results with depletion of known V-ATPase components (New Figure 4).

3) The role of TMEM199 and CCDC115 in lysosomal degradation is also supported by the increase in NCOA4 protein levels in TMEM199 or CCDC115 deficient cells. We show that NCOA4, the autophagy cargo receptor, accumulates in TMEM199 or CCDC115 depleted HeLa cells, similarly to depletion of core V-ATPase subunits or BafA treatment (New Figure 8).

In combination, these studies of endosomal acidification and lysosomal degradation strongly support a role for TMEM199 and CCDC115 in V-ATPase function. The genetic complementation studies suggested by this reviewer were also a valid suggestion, which we had already tried. However, neither TMEM199 nor CCDC115 complemented the yeast Vma12/Vma22 mutant phenotypes, and Vma12/22 did not prevent HIF1α stabilisation in the human TMEM199 or CCDC115 deficient cells. This is perhaps not surprising, given that the sequence identities are low.

*A second concern is that the authors attribute the loss of PHD activity to a loss of iron uptake in the V-ATPase-deficient cells. Cytosolic iron levels are maintained by uptake, yes, but also by the balance between storage and mobilization of iron in ferritin. Ferritin is turned over in the lysosome by an autophagic process that requires the cargo receptor NCOA4 and an appropriately acidified lysosome. While the authors show that increased iron uptake after external iron supplementation will restore cytosolic iron levels and lead to PHD activation, the defect associated with the V-ATPase may not be defective iron uptake, but defective iron recycling from ferritin. The rapid onset of PHD inactivation, even in cells pre-loaded with iron (Figure 6), suggests this may be the case. Drugs that inhibit the lysosomal V-ATPase, e. g., bafilomycin A, will also inhibit the acidification of the endosome, so these drugs don't really distinguish between impaired uptake of iron and impaired turnover of ferritin. The authors should clarify this experimentally. Iron uptake may be measured directly in assays using stable or radioactive isotopes of iron. Although direct measurement of cytosolic iron levels is technically fraught, as the authors explain, indirect measurements are possible, however. Iron Regulatory Protein 2 is rapidly degraded via the proteasome in iron-replete cells and is a good indicator of cytosolic iron availability. If the author's iron hypothesis is correct, IRP2 levels should increase with inhibition of the V-ATPase. Proteins involved in autophagic delivery of ferritin to the lysosome could be depleted and PHD activity measured. This approach would add additional clarity to the mechanism of iron depletion. As this is the most novel aspect of these studies, it represents an important opportunity for novel and significant observations.*

*The concerns I outlined above are important, because the role of the two new proteins as assembly factors is far from certain and they should present stronger evidence using more than one experimental approach to support their claim. My second concern is equally important. Investigators in the iron field have not focused on the regulation of ferritin turnover until very recently because the genetic components that carry out the turnover have only recently been identified. The role of ferritin turnover in the lysosome is the potentially novel finding in this study. We already know that PHD is an iron-dependent enzyme and that chelators that block iron uptake will lead to inactivation of PHD. We already know that drugs that inhibit the V-ATPase will inhibit the uptake of Tf-bound iron, but this effect is kind of small in cultured cells unless you treat for many hours. What we don't know is the relative importance of the flow of iron through ferritin and the turnover of ferritin in the lysosome in maintaining cytosolic iron levels and the activity of cytosolic iron enzymes. They should take a look at this before they ascribe their effects to impaired uptake. If the authors choose not to do further studies, then they must acknowledge the ambiguity and inconclusive nature of their results. Furthermore, the impact and significance of these studies will be lessened.*

The relationship between V-ATPase biogenesis and function with regard to iron metabolism is an important point that we were keen to address (see also comments to reviewer 1). The focus on iron uptake as the sole mechanism for HIF1α stabilisation following V-ATPase inhibition was not intended, particularly as the conversion of ferric to ferrous iron requires acidification of endosomes (e.g. PMID 21788477 and 16227996), and we should have made this clearer in the manuscript. Kozik et al. already provide evidence for BafA and siRNA-depletion of V-ATPase preventing iron/transferrin uptake and recycling of the transferrin receptor (PMID 23263279). However, as the V-ATPase is not only involved in iron uptake, we now include several new experiments that confirm TMEM199 or CCCDC115 depletion leads to decreased intracellular iron levels:

1) We demonstrate that Tfnr-phl is visualised in non-acidified endosomal compartments (New Figure 5 and Figure 5—figure supplement 1). As low pH is required for iron release from the transferrin receptor and conversion to the ferrous form, these experiments are consistent with V-ATPase inhibition leading to cytosolic iron depletion.

2) We have measured IRP2 (IREB2), NCOA4 and ferritin levels following BafA treatment or depletion of V-ATPase subunits to confirm that cytosolic iron is reduced and to explore ferritin recycling (New Figure 8). IRP2 levels increase following BafA treatment, similarly to levels observed with DFO, confirming that V-ATPase inhibition causes cytosolic iron depletion. NCOA4 markedly accumulates following BafA treatment, consistent with a block in autophagic lysosomal degradation. However, unlike DFO treatment, where NCOA4 increases and ferritin is degraded, BafA treatment does not affect ferritin levels, as autophagy is blocked. We also show that depletion of either known V-ATPase subunits or TMEM199/CCDC115 increase IRP2 expression and NCOA4 similarly to BafA treatment, providing further evidence that disrupting the V-ATPase results in cytosolic iron depletion (New Figure 8).

3) To further delineate the importance of iron uptake compared to recycling we have depleted several key components of iron homeostasis (NCOA4, IRP2 and the transferrin receptor) and measured their ability to stabilise HIF1α (New Figure 10). Mixed populations of transferrin receptor KOs markedly increased HIF1α levels, with IRP2 depletion having a modest effect. Thus, decreasing iron uptake or disrupting iron sensing can lead to intracellular iron depletion. However, no change in HIF1α levels were observed in the NCOA4 KOs at various time points post transduction (New Figure 10). Interestingly, NCOA4 depletion did increase the levels of IRP2, but transferrin receptor and ferritin levels did not change, suggesting that upregulating iron flux through the endosomal pathway is sufficient to maintain intracellular iron levels when NCOA4 is depleted (New Figure 10).

4) We also examined the effect of iron supplementation on mixed populations of transferrin receptor null cells. Iron treatment restored HIF1α turnover (New Figure 10), suggesting that iron can be taken up from cells by a transferrin independent mechanism. Defining the nature of this transferrin independent mechanism is beyond the scope of this manuscript, but it is likely to involve transporters such as Zip14, which are known to be involved in non-transferrin-bound iron uptake (PMID 16950869) (Discussion, fourth paragraph).

Together, these new findings suggest that decreasing iron uptake and release from acidic endosomes is the dominant mechanism for HIF1α stabilisation following V-ATPase inhibition, although there is a contribution of NCOA4 to maintaining iron homeostasis by activating IRP2 (Discussion, fourth paragraph). We believe these additional experiments on iron metabolism have significantly strengthened the manuscript.

*Reviewer #3:*

*Degradative pathways of HIF have been well studied, however precise mechanisms and role in hypoxia mediated cell effects are still not clear. The major pathway by which HIF is regulated is through a proteasomal degradation pathway initiated by PHD-dependent hydroxylation of HIF. However, it has been shown that altering lysosomal and chaperone-mediated autophagy can also lead to HIF activation. The present work through a novel forward genetic screen identified several lysosomal V-ATPases that lead to increased HIF-1α expression. Interestingly, this is not via direct lysosomal degradation pathway that was previously shown. The strength of the manuscript is the novel mechanism downstream of lysosomal function. The authors show that HIF-1α stabilization following V-ATPase disruption or inhibition is via changes in cellular iron levels. It is clear that several stimuli and several different mechanisms can alter HIF-1α expression and activity. This work provides interesting downstream mechanism from lysosomes, but understanding the physiological or pathological context of this pathway in HIF-1α activation is missing. Along the same lines, as the author's state there are several diseases due to V-ATPase mutations, some experimental data would be important to show that the initiation or progression of these disease are related to iron handling and/or HIF-1α activation.*

We were pleased that this reviewer found our mechanistic studies interesting and novel. We agree that exploring the physiological and pathological role of the V-ATPase in iron metabolism and HIF signalling will be important, and forms the basis of our future studies. However, this work will likely involve establishing transgenic animal models and is beyond the scope of this manuscript. Without such models, we can only speculate as to the importance of iron handling and HIFs in the disease process, which we now make clear in the revision. Interestingly, some existing data does already support a role for TMEM199 and CCDC115 in iron metabolism, as patients with these mutations have impaired transferrin glycosylation. We have included this point in the revised text (Discussion, last paragraph).

*The authors rule out CMA pathways due to no effect following disruption of HSC70 and LAMP2A. However, HSC70 disruption is not efficient. Similarly, LAMP2A disruption is not complete and it is not clear if this is specific for LAMP2A or are all LAMP2 isoforms reduced. The authors must demonstrate that indeed CMA is reduced by showing that degradation of well-characterized targets of CMA are altered.*

The sgRNA were designed to target all forms of LAMP2 but the antibody only recognises LAMP2A. We have also now used siRNA to confirm that HSC70 depletion does not lead to HIF1α stabilisation (New Figure 7). However, we apologise that we did not make our points sufficiently clear regarding CMA. Depletion of HSC70 and LAMP2 suggested that CMA is not involved in HIF1α degradation but confirmation that autophagy/lysosomal degradation is not responsible for HIF1α stabilisation is demonstrated by the BafA and iron supplementation experiments.

If any autophagy pathway were responsible for the stabilisation of HIF1α following V-ATPase inhibition, HIF1α turnover would not be rescued by iron treatment. To clarify this point we show that EGFR degradation is still impaired following BafA treatment, irrespective of iron supplementation (New Figure 9). Furthermore, NCOA4, an autophagy cargo receptor, still accumulates in BafA and iron treated cells (New Figure 9, Figure 10). Together, these findings providing clear evidence that autophagy is still blocked following BafA and iron treatment, and therefore CMA or any other pathway leading to lysosomal degradation cannot be responsible for the restoration of HIF1α degradation we observe. This has been clarified in the main text (Discussion).

*The authors state, "HIF activation and V-ATPase activity serves as a feedback mechanism to control iron uptake, particularly as HIFs promote iron uptake under conditions of iron deficiency". HIF-2α is the major isoform that leads to iron uptake, if this pathway is an important feedback mechanism for cellular iron response HIF-2α expression and activity should be assessed.*

This is an important point we were keen to address. We find that HIF2α is also stabilised in HeLa cells by BafA treatment or depletion of V-ATPase subunits, similarly to HIF1α (New Figure 6). In addition, we find that Heme oxygenase 1 (HO-1, also known as HMOX1), a predominantly HIF2α target gene (PMID 19706526) that degrades heme and releases iron, is also upregulated (New Figure 6).

*Similar to what is shown following BafA in Figure 6, increase in HIF-1α target gene following ATP6V1a1, TMEM199, and CCDC115 disruption would be important to demonstrate the increase in HIF-1α is functional.*

We agree that it is important to show that HIF1α is functional following depletion of TMEM199, CCDC115 or core V-ATPase subunits, and now include data showing the well validated HIF1 target, Carbonic Anhydrase 9, is induced (New Figure 6).